# HYBRID DEEP SEARCHER: SCALABLE PARALLEL AND SEQUENTIAL SEARCH REASONING

**Dayoon Ko**[1,2*] **Jihyuk Kim**[2] **Haeju Park**[2] **Sohyeon Kim**[1] **Dahyun Lee**[2] **Yongrae Jo**[2]
**Gunhee Kim**[1] **Moontae Lee**[2,3] **Kyungjae Lee**[2,4†]

[1]Seoul National University  [2]LG AI Research  [3]University of Illinois Chicago  [4]University of Seoul

dayoon.ko@vision.snu.ac.kr

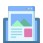 https://hybriddeepsearcher.github.io/

## ABSTRACT

Large reasoning models (LRMs) combined with retrieval-augmented generation (RAG) have enabled deep research agents capable of multi-step reasoning with external knowledge retrieval. However, we find that existing approaches rarely demonstrate *test-time search scaling*. Methods that extend reasoning through single-query sequential search suffer from limited evidence coverage, while approaches that generate multiple independent queries per step often lack structured aggregation, hindering deeper sequential reasoning. We propose a *hybrid* search strategy to address these limitations. We introduce **HybridDeepSearcher**, a structured search agent that integrates parallel query expansion with explicit evidence aggregation before advancing to deeper sequential reasoning. To supervise this behavior, we introduce **HDS-QA**, a novel dataset that guides models to combine broad parallel search with structured aggregation through supervised reasoning–query–retrieval trajectories containing parallel sub-queries. Across five benchmarks, HybridDeepSearcher significantly outperforms the state-of-the-art, improving F1 scores by +15.9 on FanOutQA and +9.2 on BrowseComp[†]. Further analysis shows its consistent test-time search scaling: performance improves as additional search turns or calls are allowed, while competing methods plateau.

## 1 INTRODUCTION

Large reasoning models (LRMs), such as OpenAI o3 (OpenAI, 2025) and DeepSeek-R1 (Guo et al., 2025), have demonstrated that allocating additional computation at inference time, commonly referred to as *test-time scaling*, can substantially improve performance. By generating longer reasoning traces through additional tokens, these models achieve stronger results on complex tasks.

Following this line of work, recent studies (Li et al., 2025; Jin et al., 2025; Zheng et al., 2025; Gao et al., 2025a) have developed retrieval-augmented generation (RAG) into deep search agents by incorporating multiple retrieval turns into the reasoning process. These agents operate in an iterative loop that issues one or more queries, retrieves relevant information, and integrates the retrieved content into subsequent reasoning steps.

However, we observe that simply extending the search chain does not yield proportional performance gains as information requirements increase. This limitation is illustrated in Figure 1, where we evaluate the test-time scaling behavior of existing search agents on BrowseComp[†]. For instance, Search-R1 (Jin et al., 2025), trained with GRPO (Shao et al., 2024) for sequential search with one query per turn, performs about 4.7 turns on average but achieves limited performance. Because it issues only a single query at each step, the total number of search calls remains around 4.6, which restricts comprehensive document exploration. Conversely, RAG-R1 (Tan et al., 2025), trained with GRPO for parallel search with multiple queries per turn, issues more queries overall (about 8.2 search calls) and gathers more information. However, it still uses only about 4.3 turns on average and achieves only marginal improvement over the sequential search baseline.

---

*Work was done during internship at LG AI Research.    †Corresponding author.

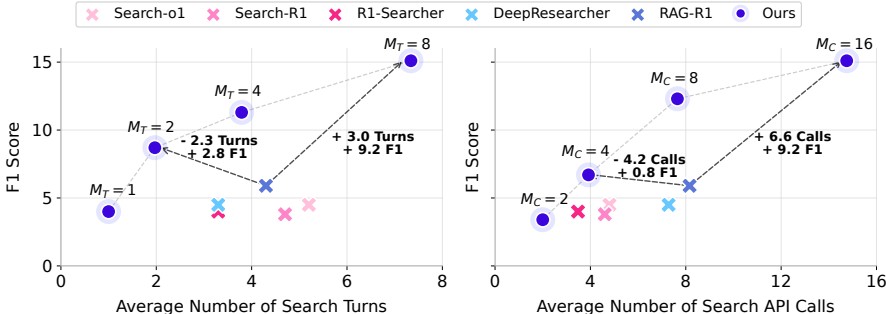

Figure 1: **Test-time Search Scaling on BrowseComp**[†]. For our method, evaluation is conducted by scaling two types of search resources: (1) latency measured by the maximum number of search turns ($M_T = 1, 2, 4, 8$), and (2) search cost measured by the maximum number of search calls ($M_C = 2, 4, 8, 16$). The x-axes report the average number of turns/calls actually used under each budget. Our model is required to output a final answer once either resource limit is exhausted. For other baselines, we allow a maximum of 10 turns with unlimited API call limits. The results on the other benchmarks are provided in A.

These results suggest that existing approaches, although designed to scale search during reasoning, fail to achieve consistent *test-time search scaling*. Specifically, models trained to extend reasoning through additional sequential search steps encounter an information bottleneck because each step retrieves only a single piece of evidence, limiting coverage growth. Models trained to generate multiple parallel queries per turn suffer from synthesis difficulty because they have not been trained to aggregate large sets of retrieved evidence without structured reasoning, often leading to premature termination. This observation raises a fundamental question: what search strategy enables genuine test-time search scaling?

In this work, we propose *hybrid* search to enable test-time search scaling. For scalable search, the search agent should (i) generate multiple queries at each step to close knowledge gaps without introducing an information bottleneck, and (ii) explicitly aggregate the retrieved evidence for subsequent sequential reasoning. Consider the question: "Out of all feature-length theatrical films directed by John Carpenter before 2015, which has the longest running time?" Once the filmography is identified, a search agent needs to retrieve all runtimes and comparing them globally. This naturally calls for parallel query expansion to collect the runtimes in one step, followed by aggregation to determine the maximum.

To instill this structured behavior, we introduce **HDS-QA**, a novel training dataset that teaches models to (i) generate broad parallel queries and (ii) explicitly aggregate results before continuing sequential reasoning. While prior approaches rely on outcome-reward online reinforcement learning with substantial computational resources, we demonstrate that direct supervision of this hybrid search strategy alone enables true test-time search scaling.

To the best of our knowledge, this is the first dataset that (i) *increases the breadth of parallel search by supporting beyond two parallel sub-queries*, and (ii) *explicitly incorporates these broad parallel search results into sequential search reasoning*. This enables scalable search through structured parallel-to-sequential reasoning. We generate these questions through a carefully designed automatic pipeline and curate answer trajectories in the form of reasoning–query–retrieval loops that include parallel search queries, resulting in 2,111 question-answer pairs.

We fine-tune an LRM on HDS-QA to build **HybridDeepSearcher**, which demonstrates consistent and monotonic search scaling as shown in Figure 1. To assess this capability, we scale two test-time search resources: (i) search turns from one to eight and (ii) search calls from two to sixteen. F1 improves by $3.8\times$ (from 4.0 to 15.1) when increasing the turn limit from $M_T{=}1$ to $M_T{=}8$, and by $4.4\times$ (from 3.4 to 15.1) when increasing the call limit from $M_C{=}2$ to $M_C{=}16$. In contrast, existing baselines plateau despite comparable or greater search budgets. HybridDeepSearcher achieves up to a threefold improvement over the state-of-the-art under matched resource constraints.

Additionally, experimental results reveal three key findings: First, it significantly outperforms all baselines *across all five benchmarks*, doubling accuracy on FanOutQA (7 evidence on average). Second, across all benchmarks, it consistently improves as search turns or calls increase, while other baselines remain stagnant or even fail to improve. Third, as the number of required evidence increases, our model shows minimal performance loss, while others suffer from a significant decline.

In summary, our contributions are as follows:

- We introduce **HDS-QA**, a novel dataset that teaches models to integrate broad parallel search with structured aggregation to enable scalable sequential reasoning.

- We propose **HybridDeepSearcher**, a structured search agent trained on HDS-QA that integrates parallel query expansion with explicit aggregation, enabling scalable and efficient search.

- **HybridDeepSearcher** exhibits test-time search scaling, improved search efficiency, broader evidence coverage, and robustness to increasing evidence requirements.

## 2 RELATED WORK

**Sequential vs. Parallel Search.** Iterative *sequential search* (Trivedi et al., 2023; Yao et al., 2023; Shao et al., 2023) has been effective for early MHQA with predefined linear reasoning paths, where a question is decomposed into interdependent sub-questions processed sequentially. For instance, IRCoT (Trivedi et al., 2023) iteratively generates a chain-of-thought sentence based on retrieved documents and performs subsequent retrieval using the sentence as a query.

Recent work (Li et al., 2025; Jin et al., 2025; Song et al., 2025; Chen et al., 2025a) has developed search agents integrating LRMs with RAG to orchestrate multi-step reasoning with external retrieval. Search-o1 (Li et al., 2025) introduces a prompt-based agentic RAG framework leveraging Reason-in-Documents for inline synthesis, while Search-R1 (Jin et al., 2025) and DeepResearcher (Zheng et al., 2025) use Group Relative Policy Optimization (GRPO) (Shao et al., 2024) to incentivize enhanced search and reasoning. However, these approaches emphasize scaling reasoning via RL while leaving search scaling largely unaddressed.

Concurrent works (Tan et al., 2025; Zhao et al., 2025) explore multi-query generation, but they are trained on questions that resort solely to either sequential or parallel search. Our contribution is the *integration of broader parallel search into sequential search reasoning to scale search*.

**Task Decomposition.** The decomposition of parallel and sequential search is closely related to task decomposition. For MHQA tasks, GenDec (Wu et al., 2024) decomposes questions into sub-queries, while Plan*RAG (Verma et al., 2025) constructs directed acyclic graphs of sub-queries. However, both methods are static and cannot adapt to intermediate retrieval results, often leading to incomplete evidence coverage.

Beyond static methods, several approaches (Zhu et al., 2024a; Prasad et al., 2024; Lee & Kim, 2023) explore dynamic decomposition across various tasks, such as web navigation. ReDel (Zhu et al., 2024a) implements a recursive multi-agent framework in which agents decompose tasks and delegate sub-tasks on the fly, producing both parallel and sequential sub-tasks. Similarly, ADaPT (Prasad et al., 2024) generates an initial plan and invokes an external verifier to trigger further hierarchical decomposition when the plan fails. These methods employ prompt-based strategies with proprietary large language models (LLMs), such as GPT-4.

These works primarily focus on how to decompose a given task effectively. However, it is equally crucial to effectively synthesize the results obtained from decomposed queries for subsequent retrieval steps in search scaling. Our dataset addresses *both decomposition and synthesis* by integrating parallel search with sequential search reasoning.

**Question Answering Datasets.** In the early stages of MHQA research, datasets such as HotpotQA (Yang et al., 2018) and 2WikiMultiHopQA (Ho et al., 2020) were widely used to train and evaluate the retrieval and reasoning capabilities of LLMs. As models have advanced, more challenging benchmarks have emerged to test increasingly complex reasoning over broader evidence coverage. MuSiQue (Trivedi et al., 2022) increases sequential complexity by chaining single-hop questions, extending reasoning from two to four hops. FanOutQA (Zhu et al., 2024b) evaluates fan-out style questions that require simultaneous retrieval across multiple independent entities.

More recently, FRAMES (Krishna et al., 2025) has been proposed to evaluate factual accuracy, retrieval ability, and reasoning in generating final answers, while BrowseComp (Wei et al., 2025) poses complex questions that demand integrating multiple factual pieces that are often difficult to locate on the web. These benchmarks reflect the growing complexity of evaluation tasks.

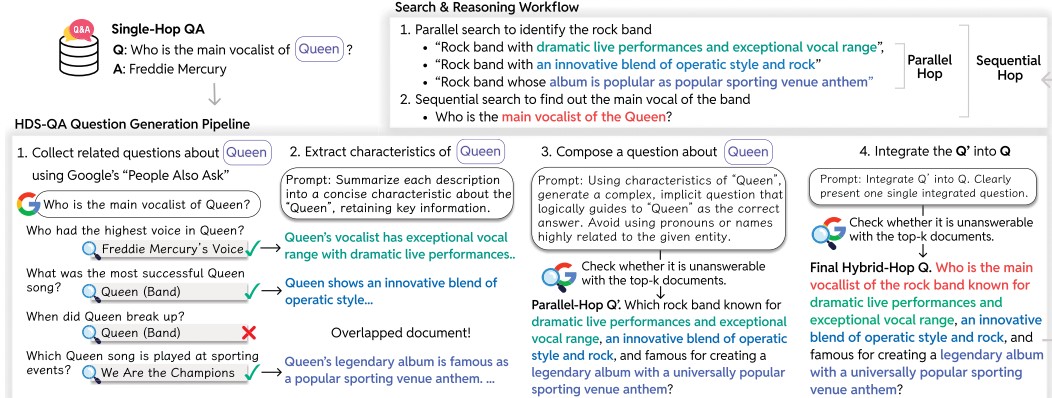

Figure 2: Pipeline for HDS-QA question generation.

Compared to recent benchmarks, progress on training datasets (*e.g.*, HotpotQA) has lagged behind in the number of hops and required evidence (at most two), leaving models unable to keep pace with increasingly complex tasks that demand processing numerous pieces of retrieved information. Concurrent synthetic-data efforts (Wu et al., 2025; Gao et al., 2025b; Liu et al., 2025) also increase task complexity, but they primarily construct questions centered on deeper or longer sequential browsing trajectories. In contrast, our training dataset is specifically designed to provide questions that involve (i) *a greater breadth* of parallel sub-queries and (ii) *seamless incorporation* of parallel search results into subsequent sequential search steps. Such explicit parallel–sequential decomposition is currently underrepresented in existing training datasets, and HDS-QA directly fills this gap.

## 3 APPROACH

We propose **HybridDeepSearcher**, an LRM capable of adaptively handling both parallel-hop and sequential-hop search strategies. In parallel-hop queries, multiple searches can be executed simultaneously without interdependence, whereas sequential-hop queries require step-by-step execution, where each query depends on the result of the previous one. To train the model for such flexible search reasoning, we introduce a novel supervised dataset, **HDS-QA**.

### 3.1 HDS-QA

HDS-QA provides complex questions that require both parallel- and sequential-hop reasoning, along with iterative reasoning-querying-retrieval trajectories to derive the correct answer for each question, enabling supervised training. All prompts used are presented in Appendix E.

**Question Generation.** As illustrated in Figure 2, our question-generation pipeline involves four key steps. We use Qwen3-32B (Yang et al., 2025) across all processes in generating questions.

1. **Entity extraction and related question collection:** Starting from a single-hop seed NQ question Kwiatkowski et al. (2019) (*e.g.*, *Who is the main vocalist of Queen?*), we extract a central entity (*e.g.*, *Queen*) via prompting. We then query Google's People Also Ask feature using the seed question to collect multiple related questions about the entity. To ensure diversity, we select only the queries that retrieve distinct top-ranked documents. As shown in the Figure 2, the related question "*When did Queen break up*" is not adopted since it retrieves the same document as "*What was the most successful Queen song?*"

2. **Entity characteristic summarization:** We summarize the retrieved documents for each related question into concise statements (three to five) representing the entity's key characteristics. We leverage the prompt for the Reason-in-Document module from Search-o1 (Li et al., 2025).

3. **Parallel-hop question formulation:** Using these characteristics, we compose a parallel-hop question implicitly referencing the entity. We prompt the model to avoid explicitly mentioning entities closely associated with the central entity, ensuring the necessity for parallel hops.

4. **Integration into hybrid-hop questions:** Finally, we replace the entity in the seed single-hop question with the parallel-hop question, introducing an additional sequential hop. To ensure that both parallel and sequential hops are genuinely required, we verify that neither the parallel-hop question nor the final hybrid-hop question can be directly answered from a single retrieval step.

Figure 2 also illustrates the workflow for solving the example question. In this example, the model is supposed to perform sequential reasoning in two steps (sequential-hops): (i) identifying the rock band and (ii) finding its main vocalist. During the first step, identifying the rock band necessitates issuing three simultaneous queries (parallel-hops). Following this pipeline, we generate a total of 1,987 hybrid-hop questions.

**Answer-trajectory Generation.** We create answer trajectories through iterative loops of reasoning, querying, and retrieval. Inspired by the prompting strategy of Li et al. (2025), we prompt the Qwen3-32B model to iteratively perform reasoning-querying-retrieval steps, emitting multiple parallelizable queries simultaneously during each querying step until a final answer is produced. In the prompt, we include a carefully designed demonstration of an answer trajectory based on the question in Figure 2. We retain a trajectory in the dataset only if its final answer is correct. Importantly, a trajectory may still contain incorrect steps, but if it ultimately recovers and produces the correct answer, it can provide valuable supervision by demonstrating how to recover from errors.

To ensure diversity in trajectories, we perform inference four times for each question and retain all successful trajectories. This allows the model to learn various valid reasoning traces and adapt flexibly based on retrieved search results. From 7,948 total attempts (1,987 questions × 4 inferences), we collect 2,111 successful trajectories. At the question level, 773 of the 1,987 questions receive at least one correct answer across their four attempts, resulting in a pass@4 of 38.9% (773 / 1,987).

Although the HDS-QA question generation pipeline is fixed, it provides a controlled and reliable setup that explicitly requires both parallel and sequential reasoning. As shown in Appendix B, the resulting HDS-QA trajectories exhibit substantial diversity in both sequential depth and parallel width, indicating highly varied emergent reasoning behaviors. This suggests that HDS-QA questions function as a stable testbed in which the model must dynamically determine when to invoke parallel retrieval and when to engage in sequential reasoning.

## 3.2 HYBRIDDEEPSEARCHER

We fine-tune Qwen3-8B on HDS-QA to create the HybridDeepSearcher model. This section describes how HybridDeepSearcher operates through iterative reasoning, querying, and retrieval steps, as shown in Table 1.

**Reasoning.** The model begins by reasoning within the $\langle\text{think}\rangle$ and $\langle/\text{think}\rangle$ tokens.

**Querying.** Based on the reasoning outputs, the model generates sequential or parallel queries within the $\langle|\text{begin\_search\_queries}|\rangle$ and $\langle|\text{end\_search\_queries}|\rangle$ tokens, separated by ";\n".

**Retrieval.** Each query is executed via a web search API, and the retrieved documents are summarized using an external model through an API call. Following Li et al. (2025) and Zheng et al. (2025), we employ a summarizer because web search results are highly noisy and often contain large amounts of irrelevant or duplicated content. Specifically, we adopt the summarization prompt from Li et al. (2025). For each query $q_i$, the summarizer produces a summary $s_i$, and we concatenate them in the form of "$q_i$: $s_i$". All such query-summary pairs are joined with newline separators,

| Example |
|---|
| **Q.** Out of all of the feature-length theatrical films that John Carpenter directed before 2015, which has the longest running time? |
| **HybridDeepSearcher's Response** |
| $\langle\text{think}\rangle$ The goal is to... I'll start by searching for a list of John Carpenter's feature films before 2015. $\langle/\text{think}\rangle$ |
| $\langle|\text{begin\_search\_queries}|\rangle$ John Carpenter's feature films before 2015;\n List of John Carpenter...\n $\langle|\text{end\_search\_queries}|\rangle$ |
| $\langle|\text{begin\_search\_results}|\rangle$ John Carpenter's feature films before 2015: John Carpenter's ...\n List of John... $\langle|\text{end\_search\_results}|\rangle$ |
| $\langle\text{think}\rangle$ ... the running time of each film and identify the longest one. I'll search for the running times of these films. $\langle/\text{think}\rangle$ |
| $\langle|\text{begin\_search\_queries}|\rangle$ Running time of Dark Star;\n... Running time of Ghosts of Mars $\langle|\text{end\_search\_queries}|\rangle$ |
| $\langle|\text{begin\_search\_results}|\rangle$ Running time of Dark Star: The running ... \n Running time of Ghosts of ... $\langle|\text{end\_search\_results}|\rangle$ |
| $\langle\text{think}\rangle$ ... From this list, the longest running time is 115 minutes for Starman (1984). $\langle/\text{think}\rangle$ |
| **Final Answer:** Starman (1984) |

Table 1: An example of HybridDeepSearcher on FRAMES.

and the final aggregated result is wrapped between the special tokens $\langle|\text{begin\_search\_results}|\rangle$ and $\langle|\text{end\_search\_results}|\rangle$ tokens.

Afterward, the model resumes reasoning, and may repeat additional querying-retrieval cycles. Once sufficient information is gathered, it produces a final answer.

## 4 EXPERIMENTAL SETUP

We evaluate our approach in both sequential and parallel search reasoning tasks, comparing its performance with several baseline models.

**Datasets.** We evaluate our method on five QA benchmarks, covering both sequential and parallel search reasoning scenarios:

- **MuSiQue** (Trivedi et al., 2022): Consists of questions generated by chaining multiple single-hop questions. Sequential hops range from 2 to 4, with some questions containing at most two parallel hops.

- **FanOutQA** (Zhu et al., 2024b): Contains fan-out style questions requiring the model to identify entities and aggregate extensive information across many documents.

- **FRAMES** (Krishna et al., 2024): Evaluates complex multi-hop retrieval strategies as well as the model's factuality and reasoning capabilities, requiring the integration of information from multiple sources.

- **MedBrowseComp** (Chen et al., 2025b): Features medical fact-seeking tasks with web browsing to deliver concise, verifiable answers, simulating real-world medical research scenarios.

- **BrowseComp** (Wei et al., 2025): Assesses the model's persistence in searching, collecting, and verifying information with inverted and complex questions, which are difficult to resolve but easy to verify. As many BrowseComp questions require exhaustive browsing, we curated a practical yet challenging subset of 150 questions (**BrowseComp**†) that OpenAI o3 can solve within a five-minute web-search limit. Specifically, we ran o3 with web search and retained questions it answered correctly within five minutes, yielding 150 question–answer pairs.

**Evaluation Metrics.** To evaluate the effectiveness and efficiency of our model, we use the following metrics:

- **F1**: We report the word-level F1 score as a measure of the accuracy of model responses. For FanOutQA, we also report the BLEURT score, a learned semantic similarity metric, in accordance with the dataset's established evaluation protocol.

- **Acc** (Model judge accuracy): Accuracy assessment generated by the model. For FanOutQA, we follow the prompt provided in Zhu et al. (2024b). For other cases, we use the prompt from Zheng et al. (2025), with Qwen3-32B to perform scoring.

- **# Turn**: We report the average number of search turns per response, measuring inference latency.

- **AUC** (Area Under Accuracy–Turn Curve): Measures efficiency as the area under the accuracy–turn curve (Figure 3), capturing the trade-off between accuracy and latency. Accuracy is computed from the mean Acc over search turns, assigning 0 if a question remains unanswered. Formally, let $Q$ be the set of evaluation questions, and $T$ the maximum number of turns. For each $q_i \in Q$, define

$$s_t(q_i) = \begin{cases} \text{Acc}(q_i), & \text{if } q_i \text{ is answered within } t \text{ turns,} \\ 0, & \text{otherwise.} \end{cases}$$

  Then the AUC is

$$\text{AUC} = \frac{1}{T} \sum_{t=1}^{T} \frac{1}{|Q|} \sum_{q_i \in Q} s_t(q_i). \tag{1}$$

  Thus, higher values indicate better efficiency.

| | MuSiQue | | | | FanOutQA | | | | | FRAMES | | | | MedBrowseComp | | | | BrowseComp[†] | | | |
|---|---|---|---|---|---|---|---|---|---|---|---|---|---|---|---|---|---|---|---|---|---|
| | F1 | Acc | #Turn | AUC | F1 | BLEURT | Acc | #Turn | AUC | F1 | Acc | #Turn | AUC | F1 | Acc | #Turn | AUC | F1 | Acc | #Turn | AUC |
| *Non-iterative* | | | | | | | | | | | | | | | | | | | | | |
| Naïve Gen | 12.8 | 16.4 | - | - | 10.9 | 27.5 | 3.2 | - | - | 14.0 | 17.5 | - | - | 8.0 | 11.9 | - | - | 0.0 | 0.0 | - | - |
| Standard RAG | 15.8 | 24.8 | - | - | 20.6 | 32.1 | 5.6 | - | - | 21.9 | 30.9 | - | - | 11.3 | 16.3 | - | - | 0.8 | 0.0 | - | - |
| *Sequential Search* | | | | | | | | | | | | | | | | | | | | | |
| Search-o1 | 23.4 | 31.8 | 3.7 | 0.26 | 26.7 | 32.9 | 8.7 | 5.2 | 0.06 | 34.2 | 48.6 | 4.3 | 0.37 | 12.9 | 21.6 | 4.7 | 0.16 | 4.5 | 2.0 | 5.2 | 0.02 |
| Search-R1 | 26.6 | 29.1 | 3.2 | 0.23 | 10.1 | 23.1 | 1.2 | 4.3 | 0.01 | 27.3 | 34.8 | 4.0 | 0.25 | 18.8 | 21.6 | 4.0 | 0.16 | 3.8 | 4.7 | 4.7 | 0.03 |
| R1-Searcher | 25.1 | 28.5 | 2.7 | 0.24 | 18.8 | 30.2 | 2.5 | 3.1 | 0.02 | 16.0 | 19.0 | 2.8 | 0.15 | 15.8 | 24.4 | 3.1 | 0.20 | 4.0 | 2.0 | **3.3** | 0.02 |
| *Parallel Search* | | | | | | | | | | | | | | | | | | | | | |
| DeepResearcher | 21.7 | 23.4 | 3.4 | 0.19 | 26.4 | 35.4 | 6.45 | 3.6 | 0.05 | 28.5 | 36.6 | 3.2 | 0.30 | 14.7 | 26.1 | 4.3 | 0.20 | 4.5 | 2.7 | **3.3** | 0.02 |
| RAG-R1 | 29.7 | 32.4 | **2.1** | 0.29 | 28.2 | 36.7 | 10.0 | 1.9 | 0.09 | 35.8 | 45.6 | **2.1** | 0.41 | 19.2 | 28.2 | **2.6** | 0.24 | 5.9 | 3.3 | 4.3 | 0.02 |
| *Hybrid Search (Ours)* | | | | | | | | | | | | | | | | | | | | | |
| HybridDeepSearcher | **31.2** | **35.1** | 3.3 | **0.30** | **44.1** | **48.4** | **20.0** | 3.1 | **0.15** | **39.1** | **54.0** | 3.4 | **0.44** | 19.8 | 30.4 | 3.4 | **0.26** | **15.1** | **13.3** | 6.0 | **0.09** |
| w/ Qwen2.5-7B-Inst | 28.1 | 32.6 | 2.8 | 0.26 | 37.4 | 43.4 | 17.4 | 3.4 | 0.13 | 39.0 | 52.4 | 3.4 | 0.42 | **23.2** | **32.7** | 3.3 | 0.25 | 8.4 | 6.0 | 6.4 | 0.04 |

Table 2: Comparison of answer accuracy on MuSiQue, FanOutQA, FRAMES, MedBrowseComp, and BrowseComp[†]. Best results in each column are marked in bold. AUC represents the area under the accuracy–turn curves (Figure 3); higher values indicate greater effectiveness with fewer search turns. BrowseComp[†] is a subset of BrowseComp consisting of 150 questions that are solvable by OpenAI o3 using web search within a 5-minute limit. We use Qwen3-8B for Naïve Gen, Standard RAG, and Search-o1.

**Baselines.** We compare our method against multiple baselines, categorized into three groups:

- Non-iterative: (i) **Naïve Generation**: inference without retrieval; (ii) **Standard RAG**: retrieves documents directly based on the input question.

- Sequential search baselines: These methods implement iterative single-query strategies. (iii) **Search-o1** (Li et al., 2025): Prompt-based iterative baseline; (iv) **Search-R1** (Jin et al., 2025) and (v) **R1-Searcher** (Song et al., 2025): Trained with GRPO on single-hop (e.g., NQ) or multi-hop QA tasks (e.g., HotpotQA), using Qwen2.5-7B-Instruct (Qwen et al., 2025) as backbone.

- Parallel search baselines: These methods implement iterative multi-query strategies. (vi) **Deep-Researcher** (Zheng et al., 2025) and (vii) **RAG-R1** (Tan et al., 2025): trained with GRPO on single- and multi-hop tasks, employing Qwen2.5-7B-Instruct.

**Experimental Details.** We employ Qwen3-8B (Yang et al., 2025) for all prompt-based baselines (i, ii, iii), enabling thinking mode for these models. All iterative methods (iii-vii) are allowed up to 10 search turns, performing reasoning after each retrieval step. Queries are executed via web search using the Jina AI API.[1] To summarize retrieved documents, we utilize the Qwen3-32B model for baselines (iii) Search-o1 and (vi) DeepResearcher as well as ours. For training HybridDeepSearcher, we fine-tune Qwen3-8B on 2,111 HDS-QA question-answer trajectory pairs, randomly split into 95% training and 5% validation, for one epoch with a learning rate of 3e-5, a batch size of 32, and gradient accumulation over 32 steps. All parameters undergo fine-tuning, and we masked the tokens between search results tokens, not applying gradient updates on the search results to prevent the model from memorizing them. Further experimental details appear in Appendix A.

## 5 RESULTS

Table 2 compares HybridDeepSearcher with the baselines in terms of answer Accuracy (F1 and Acc), average number of search turns, and AUC. We also provide qualitative analyses by comparing our method with other baselines in Appendix F; please refer to it for detailed examples.

**HDS-QA enables HybridDeepSearcher to consistently achieve the best answer Accuracy across benchmarks (Table 2).** Naïve generation performs poorly, confirming that these benchmarks require external knowledge beyond what LRMs encode. Standard RAG improves slightly, but its single-pass retrieval cannot adapt to missing information during reasoning

---

[1]https://jina.ai/reader

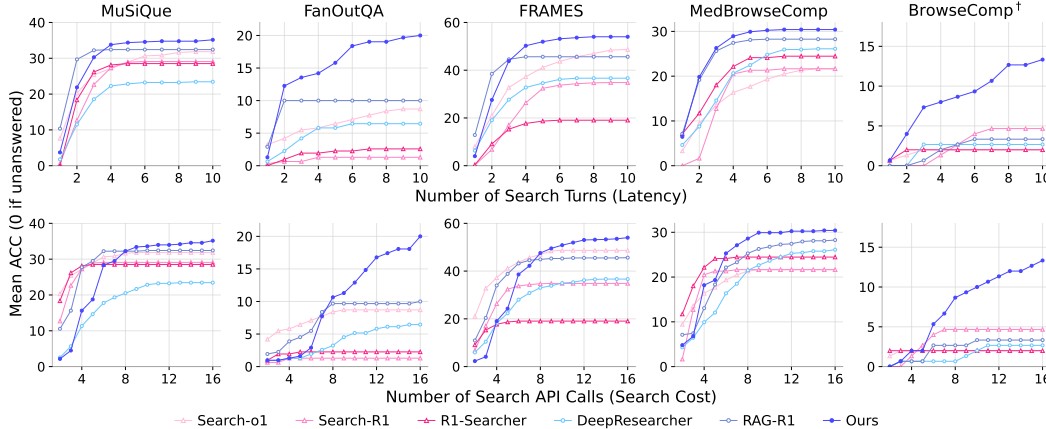

Figure 3: **Trade-off between effectiveness and efficiency.** We compare mean Acc scores by the number of search turns (upper) and search API calls (lower). At each turn or API call, we compute the mean Acc scores across all datapoints, assigning a score of 0 if unanswered within the allowed turns or calls.

Iterative single-query baselines substantially outperform standard RAG, particularly on the MuSiQue dataset, but struggle on FanOutQA and FRAMES, which require retrieving broader and more disjoint pieces of information. In these cases, multi-query baselines, DeepResearcher and RAG-R1, achieve comparable or superior Accuracy with fewer search turns (*i.e.*, lower latency). These results indicate that the ability to generate multiple queries in parallel is crucial for efficiently scaling search in scenarios requiring broader information retrieval, while iterative querying is effective in focused, narrow settings.

Nonetheless, multi-query baselines still underperform HybridDeepSearcher in both F1 and Acc, reflecting their suboptimal use of parallel search. This limitation may stem from their training data: as most are trained on HotpotQA, which involves only two sequential or parallel hops without hybrid integration. Consequently, these models show little improvement on BrowseComp[†], which demands persistent search. In contrast, HybridDeepSearcher, trained on HDS-QA with explicit hybrid supervision, consistently achieves the highest Accuracy across all benchmarks, including MedBrowseComp and BrowseComp[†], demonstrating generalizability.

For a fairer comparison, we also train Qwen2.5-7B-Instruct on HDS-QA, since all iterative search baselines except Search-o1 use it as the backbone. This model surpasses the state of the art across all benchmarks. Notably, it is trained only via supervised fine-tuning on parallel–sequential trajectories, without any RL (e.g., GRPO) for reasoning, unlike most baselines. These results suggest that scalable hybrid search behavior is learnable from supervision alone, indicating that current RL approaches such as GRPO may not provide the most effective signal for search scaling.

**HybridDeepSearcher shows a strong efficiency, balancing between effectiveness and latency (Table 2).** We introduce the AUC metric to measure the trade-off between effectiveness and latency, as noted in Eq.(1). Across all benchmarks, ours achieves the highest AUC value. Although RAG-R1 consumes significantly fewer turns to solve problems compared to other baselines, its lower performance results in a lower AUC value compared to ours. This is because RAG-R1 fails to leverage additional turns, plateauing after about 2–3 turns, as illustrated in the first row of Figure 3.

**HybridDeepSearcher scales performance with increased resource utilization (Figure 3).** Figure 3 presents mean Acc scores with respect to search turns (or search API calls), illustrating the relationship between model performance and latency (or search costs), respectively. Regarding search turns (upper), ours consistently achieves the highest Acc scores across most turns. Although RAG-R1 demonstrates better performance in the initial two turns on MuSiQue and FRAMES, it does not exhibit further improvement with additional turns. In contrast, ours progressively enhances its performance with subsequent turns. Especially on BrowseComp[†], unlike other baselines, ours consistently benefits from utilizing more turns.

In terms of API search calls (lower), ours initially shows lower performance compared to other baselines when fewer calls are utilized. Nevertheless, while other baselines reach a performance plateau after approximately eight calls, ours continues to improve performance as the number of search API calls increases, particularly on FanOutQA and BrowseComp[†]. These datasets require persistent information gathering for verification or comparison tasks, thus demanding robust search capabilities. Ours fulfills this requirement by effectively parallelizing multiple queries within fewer turns, enabling scalable query handling.

**HybridDeepSearcher significantly enhances the LRM's search capability (Table 3).** We also examine the search capability of iterative search models, a core competency of LRMs in the RAG paradigm. Specifically, we investigate whether the gold evidence documents (*i.e.*, Wikipedia links) annotated in MuSiQue, FanOutQA, and FRAMES datasets are retrieved using queries generated by models. We use the Wikimedia API to retrieve the top-10 Wikipedia links to calculate coverage. Specifically, we compute the set intersection between the gold evidence links and all retrieved links. Formally, the mean evidence coverage is calculated as follows:

$$\text{Evidence Coverage} \;=\; \frac{1}{|Q|} \sum_{q_i \in Q} \frac{|U_i \cap D_i|}{|D_i|},$$

where $q_i \in Q$ is a question in the dataset, $D_i$ is the set of gold annotated links for the $q_i$, and $U_i$ is the union of links retrieved by any of the model's queries for $q_i$. The results are reported in Table 3, where ours outperforms all the baselines across all three benchmarks. The performance gap is most pronounced in FanOutQA, which has the highest number of annotated evidence links among the three datasets. This demonstrates that ours can effectively scale the search to retrieve all necessary evidence.

**HybridDeepSearcher is more robust on questions requiring extensive evidence (Figure 4).** Figure 4 reports Acc scores grouped by the number of gold evidence documents on MuSiQue, FanOutQA, and FRAMES. We compare against Search-o1 and RAG-R1, representing strong single-query and multi-query iterative baselines. As the number of required evidence increases, questions become more challenging due to incomplete coverage. Nevertheless, HybridDeepSearcher exhibits robustness, with consistently smaller performance drops. In particular, on FRAMES, it maintains stable performance even when increasing from three to five or more evidence documents, whereas the baselines degrade significantly as evidence requirements grow. These results highlight that integrating parallel and sequential search captures both the breadth and depth of information, enabling robust scaling on complex questions.

| | Evidence Coverage Rate | | |
|---|---|---|---|
| | **MuSiQue** | **FanOutQA** | **FRAMES** |
| Search-o1 | 33.4 | 38.3 | 44.8 |
| Search-R1 | 31.6 | 39.2 | 42.2 |
| R1-Searcher | 34.2 | 35.6 | 38.6 |
| DeepResearcher | 38.8 | 49.9 | 49.0 |
| RAG-R1 | 35.9 | 53.2 | 48.0 |
| HybridDeepSearcher | **40.7** | **61.0** | **55.8** |

Table 3: Comparison of search capability with the evidence coverage rate.

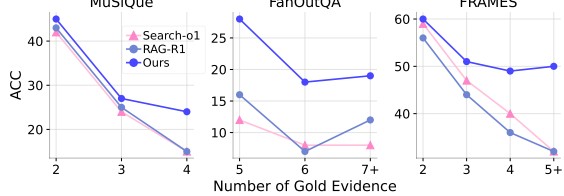

Figure 4: Acc grouped by the number of gold evidence on MuSiQue, FanOutQA, and FRAMES.

## 6 DISCUSSION

In this section, we further examine **(i)** whether the performance gain arises from hybrid search behavior rather than merely fine-tuning on HDS-QA, and **(ii)** whether reinforcement learning can further improve hybrid search.

**Does the performance gain come from HDS-QA fine-tuning or hybrid search behavior?** To isolate the effect of hybrid search, we construct an ablation that removes all parallel querying behavior. We train a model on trajectories that issue exactly one query per reasoning step while keeping the

| | MuSiQue | | | FanOutQA | | | FRAMES | | | MedBrowseComp | | | BrowseComp[†] | | |
|---|---|---|---|---|---|---|---|---|---|---|---|---|---|---|---|
| | Acc | #Turn | AUC | Acc | #Turn | AUC | Acc | #Turn | AUC | Acc | #Turn | AUC | Acc | #Turn | AUC |
| *Sequential Search* | | | | | | | | | | | | | | | |
| Search-o1 | 31.8 | 3.7 | 0.26 | 8.7 | 5.2 | 0.06 | 48.6 | 4.3 | 0.37 | 21.6 | 4.7 | 0.16 | 2.0 | 5.2 | 0.02 |
| Ours (single-query) | 21.6 | **2.9** | 0.18 | 8.7 | 4.5 | 0.06 | 33.8 | **3.2** | 0.25 | 26.8 | 4.7 | 0.10 | 6.4 | **4.3** | 0.04 |
| *Hybrid Search* | | | | | | | | | | | | | | | |
| Ours | 35.1 | 3.3 | **0.30** | 20.0 | **3.1** | **0.15** | 54.0 | 3.4 | **0.44** | 30.4 | **3.4** | **0.26** | 13.3 | 6.0 | **0.09** |
| Ours (GRPO) | **36.3** | 4.0 | 0.27 | **20.9** | 4.3 | **0.15** | **57.2** | 4.1 | 0.42 | **31.1** | 4.1 | 0.23 | **14.0** | 6.7 | 0.08 |

Table 4: Comparison of Acc, # Turn, and AUC across five benchmarks for Search-o1, our single-query fine-tuned model, HybridDeepSearcher, and HybridDeepSearcher with GRPO.

underlying questions unchanged. Using the same data-generation pipeline, this produces 2.1k valid single-query trajectories, providing a controlled comparison against HybridDeepSearcher.

As shown in Table 4, the single-query variant underperforms HybridDeepSearcher across all benchmarks, and in some cases even falls below Search-o1. These results show that the gains stem not just from exposure to the questions but, more importantly, from the hybrid search behavior itself, which improves generalization and search scalability.

**Can reinforcement learning further improve hybrid search?** Since prior work (Jin et al., 2025; Tan et al., 2025) suggests that reinforcement learning methods such as GRPO (Shao et al., 2024) can help models scale their search behavior, we investigate whether GRPO can further enhance HybridDeepSearcher. We apply GRPO on top of HybridDeepSearcher and evaluate the effects on the same set of benchmarks, using a binary correctness reward from an LLM-as-a-judge (Qwen3-4B-Instruct), indicating whether the final answer is correct.

Table 4 shows that GRPO yields modest but consistent accuracy improvements across four benchmarks. However, these gains are accompanied by increased average search depth. As a result, the AUC, which reflects both accuracy and search efficiency, decreases compared to the supervised model. Overall, these findings indicate that while GRPO can improve final answer quality, it also induces deeper search trajectories, making it less efficient than supervised hybrid-search training alone. This pattern suggests that GRPO's tendency to lengthen search trajectories does not substantially scale search, highlighting the need for more effective RL formulations for search scaling.

## 7 CONCLUSION

In this work, we address the challenge of scaling search. We propose a hybrid approach that integrates parallel and sequential search reasoning. To train models to effectively utilize this strategy, we construct **HDS-QA** via a carefully designed automatic pipeline, which generates questions that explicitly integrate broad parallel search into subsequent sequential reasoning. The dataset also includes answer trajectories represented as iterative reasoning–query–retrieval loops involving parallel sub-queries.

Through fine-tuning on HDS-QA, we develop **HybridDeepSearcher**, a model capable of seamlessly combining parallel and sequential search strategies. Experiments show that HybridDeepSearcher achieves significant performance improvements and superior efficiency as well. Further analysis demonstrates test-time search scaling, utilizing more search turns or calls for additional performance improvements, unlike all other baselines. Additionally, its sub-queries cover more evidence, resulting in a larger performance gap over the state-of-the-art on questions requiring more evidence.

Looking ahead, we plan to investigate preference optimization methods for scaling search and to extend these insights to multi-agent systems, where concurrent agents may further enhance efficiency and scalability.

## ACKNOWLEDGEMENTS

This work was supported by LG AI Research.

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

## A  EXPERIMENTAL DETAILS

### A.1  EVALUATION DATASET.

We use 512 datapoints from the MuSiQue dev set, following Zheng et al. (2025), the entire 310 datapoints from the FanOutQA dev set, all 824 datapoints from the FRAMES test set, all 605 datapoints from the MedBrowseComp evaluation set, and 150 selected datapoints from BrowseComp as described in the main text.

### A.2  COMPUTATION.

In training HybridDeepSearcher, we use eight NVIDIA A100 40GB GPUs; fine-tuning Qwen3-8B takes approximately 30 minutes. During inference, each generated query involves one Jina Search API call across all baselines and our method. Additionally, one LLM (Qwen3-32B) summarization API call is made per generated query for Search-o1, DeepResearcher, and our method. For generating LLM responses, we utilize vLLM on an A100 40GB GPU. For GRPO, we use eight H100 80 GPUs.

### A.3  HYPERPARAMETERS.

Following previous work (Li et al., 2025), we set the maximum number of search turns to 10. During inference with vLLM, we set `tensor_parallel_size` to 4, `enforce_eager` to True, `max_num_seqs` to 16, `temperature` to 0.6, and `top_p` to 0.95, following the guidelines provided in the Qwen3 technical report. For GRPO, we use a global batch size of 32, a rollout of 16 per question, and 20 steps.

### A.4  LLM USAGE

We have used LLMs to polish writing for grammar correction and rephrasing.

## B  ANALYSIS OF REASONING TRAJECTORY DIVERSITY IN HDS-QA.

This section provides additional statistics that clarify the diversity of reasoning trajectories induced by our question generation pipeline.

Our question generator adopts a minimal structural template consisting of one parallel hop and one sequential hop. While the surface form is fixed, the realized reasoning trajectories vary substantially in both: (i) **parallel width (measured per turn)** and (ii) **sequential depth (measured per question)**.

### B.1  PARALLEL RETRIEVAL WIDTH (PER TURN)

Parallel width is measured at the **turn level**. "Width" denotes the number of parallel queries issued in a single turn, and the count corresponds to the number of turns exhibiting that width.

| # Parallel Queries (Width) | 1 | 2 | 3 | 4 | 5+ |
|---|---|---|---|---|---|
| # Turns | 586 | 1958 | 2044 | 166 | 36 |

Table 5: Distribution of parallel retrieval width per turn.

These statistics show that many turns involve multi-branch retrieval (width $\geq 2$), with a substantial number requiring three parallel queries. This indicates that even under a fixed template, the system frequently expands laterally to gather multiple pieces of evidence within a single reasoning step.

## B.2 SEQUENTIAL REASONING DEPTH (PER QUESTION)

Sequential depth is measured at the **question level**. "Depth" denotes the number of sequential reasoning steps required to reach the final answer.

| Sequential Depth | 1 | 2 | 3 | 4 | 5+ |
|---|---|---|---|---|---|
| # Questions | 211 | 1319 | 424 | 119 | 38 |

Table 6: Distribution of sequential reasoning depth per question.

The distribution demonstrates meaningful variation in reasoning depth, including multi-step chains beyond three hops.

Overall, although the question template is structurally minimal, the induced reasoning trajectories exhibit substantial diversity in both lateral expansion (parallel width per turn) and longitudinal chaining (sequential depth per question). This structural diversity enables the supervised model to adaptively handle varying degrees of sequential and parallel search reasoning.

## C ADDITIONAL EXPERIMENTS

### C.1 EXTENDED ANALYSIS OF TEST-TIME SEARCH SCALING ON ADDITIONAL DATASETS

We extend the analysis of test-time search scaling (initially shown in Figure 2 of the main text) to additional datasets. The results are presented in Figures 5a and 5b. Specifically, we control two search budgets: (i) the number of search turns ($M_T \in [1, 2, 4, 8]$), and (ii) the number of search API calls ($M_C \in [2, 4, 8, 16]$). While other baselines are not constrained by these budgets, our method is required to produce a final answer once either budget is exhausted. In detail, when the number of proposed parallel queries exceeds the remaining $M_C$, we execute only the first subset of queries up to the remaining budget. Additionally, although the MedBrowseComp dataset contains unanswerable questions, we compute performance scores using only the answerable questions for fair comparisons across budget settings, as lower-budget scenarios may disproportionately benefit from the presence of unanswerable questions.

Regarding the number of search turns, our model generally achieves comparable performance even with fewer turn budgets. Although RAG-R1 slightly outperforms ours on MuSiQue and FRAMES under lower turn budgets, it does not significantly benefit from utilizing larger turn budgets. In contrast, our model effectively scales its performance with an increased number of turns, eventually surpassing RAG-R1.

In terms of the number of search API calls, our method consistently outperforms the baseline on FanOutQA and MedBrowseComp, even when using fewer API calls. However, on MuSiQue and FRAMES, our approach initially exhibits lower performance than other baselines when fewer than 8 search calls are used. Nevertheless, our method can effectively scale performance with an increased number of calls, achieving comparable or superior results—particularly when leveraging *parallel search* strategies.

Overall, these results indicate that integrating sequential and parallel search not only reduces latency and achieves competitive performance with fewer turns but also effectively scales performance when additional budget is available. This improvement arises because our method dynamically adjusts retrieval strategies and employs adaptive workflows to efficiently manage large numbers of documents for complex questions.

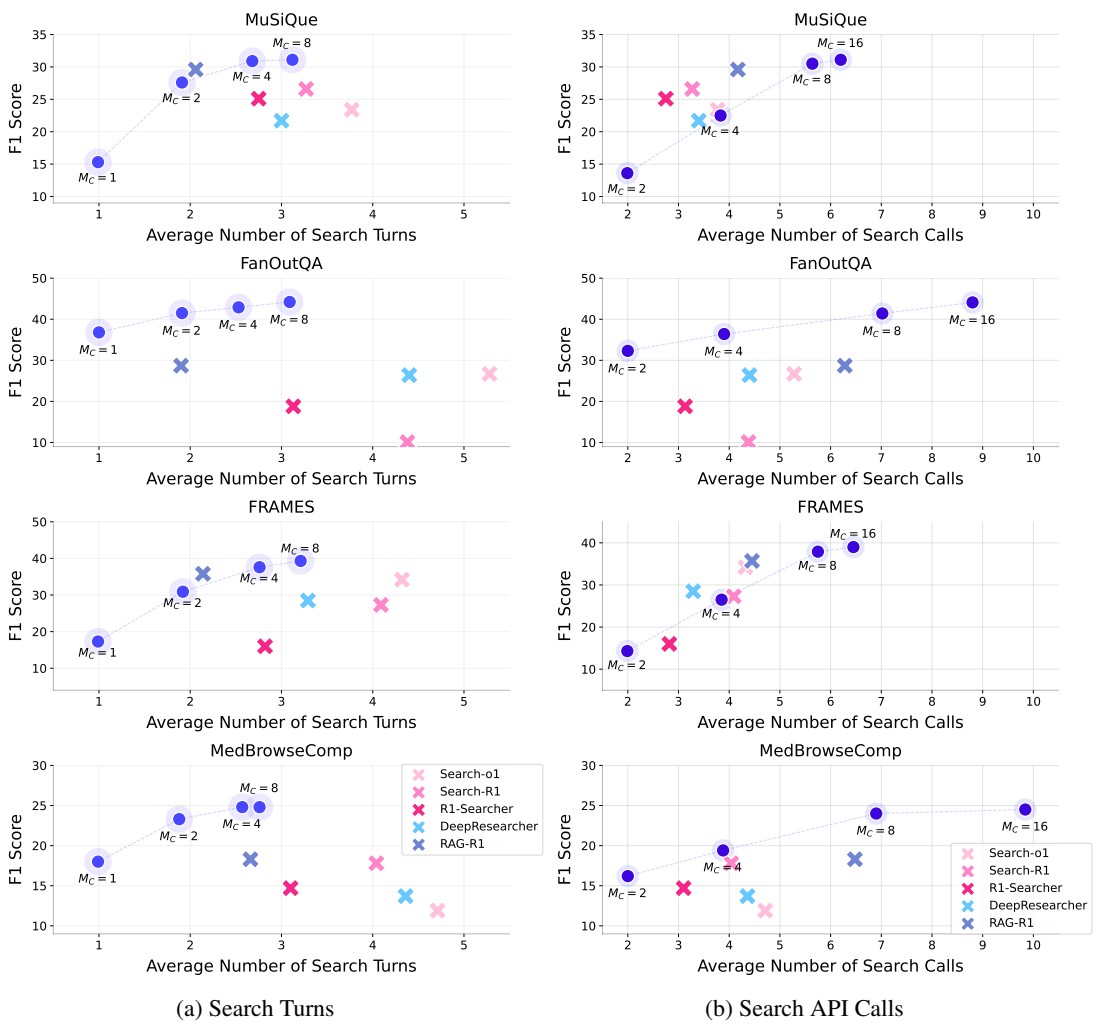

(a) Search Turns           (b) Search API Calls

Figure 5: Test-Time Search Scaling results: (a) number of turns and (b) number of API calls.

## C.2 EFFECT OF THE GENERATED-TOKEN BUDGET ON MEAN MBE SCORES

We investigate how the mean MBE score when the number of tokens the LLM generates increases. As Figure 4 in the main body, we assign 0 if unanswered within the allowed tokens. Specifically, only tokens produced by the model itself are counted; tokens originating from retrieved search snippets are excluded.

As shown in Figure 6, ours benefits consistently from a larger token budget, with especially pronounced gains on FanoutQA, BrowseComp[†]. In contrast, RAG-R1 gains almost no benefit from additional tokens, demonstrating limited scalability. Search-o1 and DeepResearcher improve as the number of generated tokens grows, but they start from a much lower baseline, indicating that they require considerably more inference cost to achieve competitive performance.

# D ABLATION STUDY

## D.1 MODEL SCALE

To examine whether our approach generalizes beyond 7–8B models, we conduct experiments using both a smaller backbone (Qwen3-4B) and a larger backbone (Qwen3-32B). We compare models fine-tuned on HDS-QA against Search-o1, and the results are summarized in Table 7.

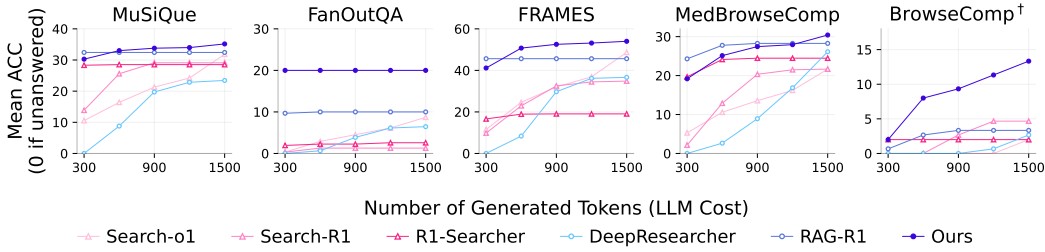

Figure 6: Comparison of Mean MBE Scores by the Number of Generated Tokens

Across both model scales, HDS-QA yields substantial gains over Search-o1. These results indicate that HDS-QA provides effective supervision by supplying coherent and reliable hybrid search traces, regardless of model size. Importantly, the hybrid-search trajectories used in these experiments were generated by the *same* 32B model, demonstrating that a teacher model larger than the student is *not* required.

| | MuSiQue | FanOutQA | FRAMES | MedBrowseComp | BrowseComp[†] |
|---|---|---|---|---|---|
| *Qwen3-4B* | | | | | |
| Search-o1 | 28.5 | 8.1 | 45.2 | 19.2 | 2.0 |
| Ours | **33.4** | **17.7** | **51.8** | **30.4** | **5.3** |
| *Qwen3-32B* | | | | | |
| Search-o1 | 35.9 | 17.7 | 58.1 | 33.2 | 6.0 |
| Ours | **37.5** | **30.0** | **59.8** | **36.0** | **15.4** |

Table 7: Performance across different model sizes.

## D.2 SUMMARIZER

To analyze the effect of a summarizer, we conduct the same experiments on HybridDeepSearcher using (i) a smaller summarizer and (ii) no summarizer. As in the main experiments, we use the `Qwen3-32B` model as a summarizer for HybridDeepSearcher, and adopt `Qwen3-8B` model as a lightweight alternative.

Table 8 shows the results. Both the 8B and no-summarizer variants outperform the strongest baseline, RAG-R1, showing only modest average drops of 1.2 and 0.7 points, respectively. These results indicate that a large 32B summarizer is not strictly necessary for strong performance. HybridDeepSearcher remains robust, even with a weaker or absent summarizer, highlighting its practicality of our approach.

| | MuSiQue | FanOutQA | FRAMES | MedBrowseComp | BrowseComp[†] |
|---|---|---|---|---|---|
| *Parallel Search* | | | | | |
| DeepResearcher | 23.4 | 6.45 | 36.6 | 26.1 | 2.0 |
| RAG-R1 | 32.4 | 10.0 | 45.6 | 28.2 | 2.0 |
| *Ours* | | | | | |
| HybridDeepSearcher | 35.1 | **20.0** | 54.0 | 30.4 | **13.3** |
| w/ 8B Summarizer | **37.6** | 16.1 | **55.6** | 29.5 | 11.9 |
| w/o Summarizer | 34.3 | 15.8 | 51.1 | **33.4** | 10.5 |

Table 8: Ablation study with a summarizer.

# E PROMPTS

**Prompt for Entity Extraction** The prompt below extracts proper nouns from a given single-hop question–answer pair to identify the central entity. These entities serve as the anchor for retrieving related questions in our dataset construction process.

---

**Prompt for Entity Extraction**

**Task Instruction:**
Identify and list all proper nouns (names of specific people, places, characters, titles, etc.) from the provided **Question** and **Answer**.

**Guidelines:**
1. **Analyze the Input:**
- Review both the question and answer carefully.
- Extract proper nouns that refer to specific entities.

2. **Output Format:**
Provide the results strictly following this JSON format:
{{
    "question": ["Proper nouns from the question"],
    "answer": ["Proper nouns from the answer"]
}}

**Example:**
Input:
Question: who does seth macfarlane play on american dad
Answer: stan smith and roger

Output:
{{
    "question": ["Seth MacFarlane", "American Dad"],
    "answer": ["Stan Smith", "Roger"]
}}

**Inputs:**
- **Question:**
{question}

- **Answer:**
{answer}

Now, extract proper nouns from the provided question-answer pair.

---

**Prompt for Documents Summarization**    Inspired by the Search-o1 Reason-in-Documents module, this prompt instructs the model to review the retrieved web pages, identify factual information relevant to each related *People Also Ask* query, and generate a clear, concise answer. The response should directly address the query and reference both the source pages and the provided reference entity for proper grounding.

---

**Prompt for Webpage Reasoning**

### Task Instruction:
You are tasked with reading and analyzing web pages based on the following inputs: **Search Query**, **Searched Web Pages**, and **Reference Entity**. Your objective is to provide sentences that directly answer the **Search Query**, using relevant information found in the **Searched Web Pages** and grounding the answer in the context of the **Reference Entity**.

### Guidelines:
1. **Analyze the Searched Web Pages:**
- Carefully review each searched web page.
- Identify the most relevant factual information to directly answer the **Search Query**.

2. **Formulate an Answer:**
- Summarize your analysis in one clear, accurate, and grammatically correct sentence that explicitly addresses the **Search Query**.
- The answer ranges from 1 to 3 sentences.
- Ensure that the answer clearly references the provided **Reference Entity**

3. **Output Format:**
- **If helpful information is found:** Present your answer in 1 to 3 sentences beginning with:
'**Final Information**'

- **If no helpful information is found:** Output the following:
'**Final Information** No helpful information found.'

### Inputs:
- **Search Query:**
{search_query}

- **Searched Web Pages:**
{document}

- **Reference Entity:**
{reference_entity}

Analyze each web page and clearly answer the query "{search_query}" in 1 to 3 sentences.

---

**Prompt for Entity Characteristics Summarization**  The prompt below further summarizes the retrieved documents' summarization about a given entity into concise statements that preserve the essential information. These summaries are intended to serve as input for generating parallel-hop questions that indirectly refer to the target entity.

---

**Prompt for Clue Summarization**

### Task Instruction:
You are given an entity and a list of clues about the entity. Your task is to summarize each clue into a concise clue about the entity, but remain the key information of the clue.

### Guidelines:

1. **Summarize Clues:**
- Summarize each clue into a concise clue.
- Remain the key information of the clue.

### Inputs:
- **Entity:**
{entity}

- **Input Clues:**
{input_list}

### Output Format:
**Summarized Clues:**
[
   "{{clue 1 summary}}",
   "{{clue 2 summary}}",
...
]

---

**Prompt for Complex Question Generation**  This prompt generates a complex, implicit question using a list of summarized clues. The question should logically lead to the target entity without explicitly naming it, enabling a parallel-hop reasoning step.

---

**Prompt for Complex Question Generation**

### Task Instruction:
You are provided with an entity and a set of clues. Then, generate a complex, implicit question that logically guides to the provided entity as the correct answer, without explicitly naming it or the related entities removed from the clues.

### Guidelines:
1. **Analyze the Clues:**
- Carefully examine each clue provided.
- Identify unique characteristics or context from these clues that indirectly lead to the given entity.

2. **Generate a Complex Question:**
- Formulate an insightful, implicit question.
- Your question should guide logically towards the entity, encouraging deduction.
- Avoid using pronouns or names in the clues that are highly related to the given entity.

### Example:

---

- **Entity:**
Queen

- **Clues:**
1. Known for energetic and theatrical live performances.
2. Freddie Mercury was famous for a wide vocal range.
3. Famous for blending rock with operatic and theatrical styles.
4. Produced the legendary album "A Night at the Opera."
5. Noted for the iconic anthem frequently performed at sports events.

- **Correct Output:**
**Complex Question:** Which celebrated rock band, recognized for energetic and theatrical live performances and a lead singer renowned for his exceptional vocal range, is famed for an innovative blend of operatic style and rock, creating a legendary album that includes a universally popular anthem commonly heard in sporting venues?
—
### Now Complete the Task:

- **Entity:**
{entity}

- **Selected Clues:**
{input_list}

### Output Format:
**Complex Question:** {{complex_question}}

**Prompt for Question Integration**    The prompt below demonstrates how to construct a hybrid-hop question by integrating a parallel-hop question into a seed single-hop question, replacing the central entity.

---

Prompt for Question Integration

**Task Instruction:**
You have two questions provided as inputs (**Q1** and **Q2**). Your task is to integrate the descriptive content of **Q2** (which answers the entity entity) into **Q1** by replacing only the specified entity (entity) in **Q1**.

**Guidelines:**

1. **Identify Entity:**
- Clearly identify the entity (entity) within Q1 to replace.

2. **Integration Procedure
- Replace only the entity (entity) from Q1 with the descriptive content of Q2.
- The result must be one cohesive, grammatically correct, and logically coherent question.
- Do not concatenate two separate questions. Instead, integrate smoothly.

3. **Output Format:**
- Clearly present one single integrated question.

**Example:**

**Inputs:**
- **Q1:** Who is the lead vocal in Queen?
- **Q2:** Which celebrated rock band, recognized for dramatic live performances and a lead singer renowned for his exceptional vocal range, is famed for an innovative blend of operatic style and rock, creating a legendary album that includes a universally popular anthem commonly heard in sporting venues? (Answer: Queen)

- **Output:**
**Integrated Question:** Who is the lead vocal in the rock band, recognized for dramatic live performances and a lead singer renowned for his exceptional vocal range, is famed for an innovative blend of operatic style and rock, creating a legendary album that includes a universally popular anthem commonly heard in sporting venues?

—
**Now Complete the Task:**
**Inputs:**
- **Q1:** {question_1}
- **Q2:** {question_2} (Answer: {entity})

**Output Format:**
**Integrated Question:** {{integrated_question}}

**Prompt for Model Response Generation**   The prompt below instructs the model to perform multi-step reasoning and search in order to assess whether a given question can be answered in a single retrieval step. It guides the model to emit search queries when needed, interpret retrieval results, and iteratively construct answer trajectories that validate the necessity of multi-hop inference. The resulting answer trajectories are used to construct our training dataset, retaining only those whose final answers are correct.

---

**Prompt for Response Generation**

### Task Instruction:
You will be given a question. Your task is to generate a detailed and step-by-step reasoning answer with parallel web search queries.

### Guidelines for Reasoning Answer Generation:
- For each step, explicitly perform the suggested queries, using:
< | begin_search_queries | >
query 1;
query 2;
...
< | end_search_queries | >
- When you have multiple queries that can be parallelized, put them together in one < | begin_search_queries | > and < | end_search_queries | > block separated by semicolons.
- The search results are followed by the search queries between
< | begin_search_results | >...< | end_search_results | >. So, clearly confirm findings at each intermediate step and state a concise conclusion before moving to the next step.
- Use natural and conversational reasoning transitions (e.g., "First", "Wait", "Next", "Alternatively") to improve readability.
- Clearly present the final answer at the end of the reasoning as follows:
**Final Answer:**
\\boxed{{YOUR_ANSWER}}

### Example

**Inputs:**
- Question:
Who is the lead vocal in the rock band recognized for dramatic live performances, a lead singer renowned for exceptional vocal range, an innovative blend of operatic style and rock, and a legendary album featuring an anthem frequently heard at sports events?

**Output:**
<think>
Okay, let's analyze the clues step by step. The question asks to find the lead vocalist of a rock band known for dramatic live performances and operatic-rock fusion, the lead vocalist's exceptional vocal range, and an universally recognized sports anthem. Let's start by searching for the rock band and then the lead vocalist.

Initially, several bands might fit this description, so I should perform some searches to narrow this down:
</think>

< | begin_search_queries | >
Rock band known for dramatic live performances and operatic-rock fusion;
Rock band with lead vocalist known for exceptional vocal range;
Rock band anthem commonly played at sports events
< | end_search_queries | >

< | begin_search_results | >

---

... The described band is Queen. ...

< | end_search_results | >

Given these strong clues, it could directly point towards Queen, particularly due to their operatic style, famously exemplified in the legendary album "A Night at the Opera". Indeed, Queen's anthems like "We Will Rock You" and "We Are the Champions" are frequently played at sporting events. Their vocalist, Freddie Mercury, is widely celebrated for his exceptional vocal range and theatrical stage presence.

To confirm this clearly, let's verify the lead vocalist of Queen explicitly:

< | begin_search_queries | >
The lead vocalist of Queen
< | end_search_queries | >

< | begin_search_results | >

... The lead vocalist of Queen is Freddie Mercury. ...

< | end_search_results | >

Given these confirmations, the band described by the clues is indeed Queen, and their iconic lead vocalist was Freddie Mercury.

**Final Answer:**
\\boxedFreddie Mercury

—

### Inputs:

- **Question:**
{question}

Now, generate your detailed reasoning answer based on the provided question.

## F  CASE STUDY

From Table 9 to Table 19, we present examples of outputs from our HybridDeepSearcher on the MuSiQue, BrowseComp, FRAMES, respectively. For each case, we selectively excerpt a portion of the raw model output to fit the page limit. The model reasoning steps are enclosed within $\langle \text{think} \rangle$ and $\langle /\text{think} \rangle$. Search queries generated by the model are enclosed within $\langle |\text{begin\_search\_queries}| \rangle$ and $\langle |\text{end\_search\_queries}| \rangle$, while the refined search results are enclosed within $\langle |\text{begin\_search\_results}| \rangle$ and $\langle |\text{end\_search\_results}| \rangle$. We observe that our trained model effectively leverages parallel querying to retrieve relevant information, enabling accurate answer generation with reduced context. In contrast, existing methods often rely on sequential querying, which results in longer contexts that hinder correct answer generation, accumulate retrieval errors, or fail to incorporate necessary constraints during the query generation stage.

Table 9: An example on **MuSiQue** dataset answered by ours, with special symbols used in the search queries and search results.

| Example from MusiQue Dataset |
|---|

**Question:**
When did the first mosque open where Federated States of Micronesia Maritime Boundary Treaty was signed?
**Answer:** September 2012

| Model Output by Ours |
|---|

⟨think⟩
Let's break down the question. The task is to find out when the first mosque opened where the Federated States of Micronesia (FSM) Maritime Boundary Treaty was signed. This requires two key pieces of information:
1. **Where was the FSM Maritime Boundary Treaty signed?**
2. **When was the first mosque built in that location?**
I'll start by determining the location of the FSM Maritime Boundary Treaty. Since this is a treaty, it was likely signed in a specific city or country. I'll search for the location of the treaty signing.
⟨/think⟩

⟨|begin_search_queries|⟩
Where was the FSM Maritime Boundary Treaty signed?;
Location of the FSM Maritime Boundary Treaty signing
⟨|end_search_queries|⟩

⟨|begin_search_results|⟩
Where was the FSM Maritime Boundary Treaty signed?: The FSM Maritime Boundary Treaty with the Marshall Islands was signed in **Majuro, Marshall Islands** on **5 July 2006**, as confirmed by the Wikipedia page on the Marshall Islands 2013Federated States of Micronesia Maritime Boundary Treaty...
⟨|end_search_results|⟩

⟨think⟩
Now that I know the FSM Maritime Boundary Treaty was signed in **Majuro, Marshall Islands**, I need to determine when the first mosque was built in Majuro. **This is a historical question, so I'll search for the construction date of the first mosque in Majuro.**
⟨/think⟩

⟨|begin_search_queries|⟩
When was the first mosque built in Majuro, Marshall Islands?;
Construction date of the first mosque in Majuro
⟨|end_search_queries|⟩

⟨|begin_search_results|⟩
When was the first mosque built in Majuro, Marshall Islands?: The first mosque in Majuro, Marshall Islands, named Bait Ul Ahad Mosque (also referred to as Baet-Ul-Ahad Mosque), was built and inaugurated in **2012**. Multiple sources confirm that the mosque was officially opened in **September 2012**, with specific mentions of **21 September 2012** as the opening date...
⟨|end_search_results|⟩

⟨think⟩
So the FSM Maritime Boundary Treaty was signed in **Majuro, Marshall Islands** on **5 July 2006**, and the first mosque in Majuro was built and inaugurated on **21 September 2012**. This means the mosque was built **after** the treaty was signed. The question asks for the date the first mosque opened where the treaty was signed, which is in Majuro. **The answer is **21 September 2012**.**
⟨/think⟩

**Final answer**: 21 September 2012

Table 10: An example on **MuSiQue** dataset answered by RAG-R1, with special symbols used in the search queries and search results.

| Example from MuSiQue Dataset |
| --- |

**Question:**
When did the first mosque open where Federated States of Micronesia Maritime Boundary Treaty was signed?
**Answer:** September 2012

| Model Output |
| --- |

⟨think⟩
To answer this question, I need to identify two key pieces of information:
1. The location where the Federated States of Micronesia Maritime Boundary Treaty was signed.
2. The date when the first mosque opened in that location.
I do not have immediate information on either of these points, so I will need to search for them.
⟨/think⟩

<search>
Federated States of Micronesia Maritime Boundary Treaty location , first mosque in Palau opening date
# Incorrectly parallelize sequentially dependent query
</search>

<information>
[{'title': 'Marshall Islands Federated States of Micronesia Maritime Boundary', 'url': 'https://en.wikipedia.org/wiki/Marshall_Islands_Federated_States_of_Micronesia_Maritime_Boundary_Treaty', 'snippet': 'The Marshall Islands Federated States of Micronesia Maritime Boundary Treaty is a 2006 treaty Location, Majuro, Marshall Islands. Parties. Marshall' . . .
</search>

⟨think⟩
Based on the search results, **the capital of Palau is Ngerulmud, but most of the population lives in Koror. The first mosque in Palau is located in Koror.** Now, I need to find the specific date when the first mosque in Koror opened.
⟨/think⟩

<search>
First mosque in Koror Palau opening date
</search>

<information>
[{'title': 'Islam in Palau - Wikipedia', 'url': 'https://en.wikipedia.org/wiki/Islam_in_Palau', 'snippet': 'Islam is a minority religion in Palau. Palau Central Jame Masjid, Koror. Contents. 1 History; 2 Demographics; 3 Islamic Society; 4 See also; 5 References' ...
} ...
</information>

⟨think⟩
The search results **did not provide a specific date for the opening of the first mosque in Koror, Palau.** However, I can infer that the first mosque in Koror is the Palau Central Jame Masjid. I will refine my search to find the exact opening date of this mosque.
⟨/think⟩

<search>
Palau Central Jame Masjid opening date
</search>

. . . (details omitted) . . .
⟨think⟩
**Unfortunately, the search results did not provide a specific opening date for the Palau Central Jame Masjid.** However, I can conclude that the first mosque in Palau, located in Koror, is the Palau Central Jame Masjid. Without a specific date, I will provide the best answer based on the available information.
**Since the treaty location is not definitively confirmed, and the mosque opening date is the most concrete piece of information, I will provide the date of the first mosque in Palau.**
⟨/think⟩

<answer> **2012** </answer>

Table 11: An example on **MuSiQue** dataset answered by DeepResearcher, with special symbols used in the search queries and search results.

| Example from MusiQue Dataset |
|---|

**Question:**
When did the first mosque open where Federated States of Micronesia Maritime Boundary Treaty was signed?
**Answer:** September 2012

| Model Output by DeepResearcher |
|---|

⟨think⟩
To answer this question, we need to find the location where the Federated States of Micronesia Maritime Boundary Treaty was signed and then find out when the first mosque opened in that location. Let's start by searching for the signing location of the treaty.
⟨/think⟩

<tool_call>
{"name": "web_search", "arguments": {"query": [ "location of Federated States of Micronesia Maritime Boundary Treaty signing" , "when was the first mosque opened in the treaty signing location" ]}}
</tool_call>

<tool_response>
[{'search_query': 'location of Federated States of Micronesia Maritime Boundary Treaty signing',
'web_page_info_list': [...] }, {'search_query': 'when was the first mosque opened in the treaty signing location',
'web_page_info_list': [...]}]
</tool_response>

⟨think⟩
**The searches did not return any relevant web pages.** The treaty signing location may not be a well-known site or might not be linked to a specific mosque. I will attempt to find the treaty signing location first.
⟨/think⟩

<tool_call>
{"name": "web_search", "arguments": {"query": [ "location of Federated States of Micronesia Maritime Boundary Treaty signing 2025" , "when was the first mosque opened in the treaty signing location 2025" ]}}
</tool_call>

...\n ...

⟨think⟩
Looking up on the relevant information, the first mosque in Nice is Masjid Algeria which was established in 1904.
Since the treaty was signed in Nice, France in June 2025 and there is historical information suggesting the first mosque in Nice, France was established in 1904, **we can conclude that the first mosque in Nice was opened before the signing of the treaty.**
⟨/think⟩

<answer> **1904** </answer>

Table 12: An example on **BrowseComp** dataset answered by ours, with special symbols used in the search queries and search results.

---

**Example from BrowseComp Dataset**

---

**Question:**
There was a global report released by the World Health Organisation after 2011 and before 2020 in which the forward was written by an Eritrean person who had attained a Master of Science (MSc) in Immunology of Infectious Diseases from a UK University. The introduction was written by someone whose first name began with Z and who wrote an opinion piece with a person whose first name began with the initials J, published after 2020 but before 2024. The cover was designed by a graphic artist who had completed a Leadership Strategies in Book Publishing somewhere after 2010 and before 2020 and had attained a Bachelor 2019s Degree in Graphic Design after 2000 and before 2010. This designer had previously worked at Malaria Consortium and Ogilvy & Mather. What is the first and last name of the cover designer of this report?

**Answer:** Cristina Ortiz

---

**Model Output by Ours**

---

⟨think⟩
Let's tackle this complex question step by step. The goal is to identify the first and last name of the cover designer of a specific WHO report. The report was released between 2011 and 2020, and the cover designer is described as someone who:
1. Completed a Bachelor 2019s Degree in Graphic Design between 2000 and 2010.
2. Worked at Malaria Consortium and Ogilvy & Mather.
3. Had a Leadership Strategies in Book Publishing course after 2010 and before 2020.
Additionally, the report's introduction was written by someone whose first name begins with Z, and the forward was written by an Eritrean person with an MSc in Immunology of Infectious Diseases from a UK university. These clues might help narrow down the report in question. First, I need to identify the WHO report that fits this timeline and the description of the cover designer. **Let's start by searching for WHO reports published between 2011 and 2020 that match the cover designer's background.**
⟨/think⟩

⟨|begin_search_queries|⟩
WHO reports between 2011 and 2020;

WHO report cover designer with Bachelor's in Graphic Design and experience at Malaria Consortium and Ogilvy & Mather;

WHO report introduction written by someone with first name Z;

WHO report forward written by Eritrean with MSc in Immunology of Infectious Diseases from UK university
⟨|end_search_queries|⟩

⟨|begin_search_results|⟩
...\n ...
WHO report cover designer with Bachelor's in Graphic Design and experience at Malaria Consortium and Ogilvy & Mather:
**Cristina Ortiz** is a graphic designer with a Bachelor's Degree in Graphic Design from EASD (2005-2009) and has worked at both Malaria Consortium (2011-2017) and Ogilvy & Mather (October 2010-2013 December 2010). She has experience in designing publications and has been associated with the World Health Organization (WHO). This information aligns with the search query for a WHO report cover designer with a Bachelor's in Graphic Design and experience at Malaria Consortium and Ogilvy & Mather.
⟨|end_search_results|⟩

⟨think⟩
Let's start by identifying the WHO report that fits the timeline and the description of the cover designer. The first clue is that the report was published between 2011 and 2020. The second clue is that the cover designer has a Bachelor's Degree in Graphic Design between 2000 and 2010, worked at Malaria Consortium and Ogilvy Mather, and has a Leadership Strategies in Book Publishing course after 2010 and before 2020.
**From the search results, we found that Cristina Ortiz** is a graphic designer with a Bachelor's Degree in Graphic Design from EASD (2005-2009), and she has worked at both Malaria Consortium (2011-2017) and Ogilvy Mather (October 2010-2013 December 2010). She also has experience in designing publications and has been associated with the World Health Organization (WHO). This aligns with the description of the cover designer.
...\n ...
Next, we need to confirm that the WHO report in question was published between 2011 and 2020 and that Cristina Ortiz was the cover designer. The report World Health Statisticsïs an annual report published by WHO, and it fits the timeline. The cover designer for the 2015 edition of World Health Statisticsïs Cristina Ortiz. This is supported by the information that she worked at Malaria Consortium and Ogilvy Mather, which aligns with her background.
**Given the information available, the most likely candidate for the cover designer is Cristina Ortiz**. She fits the description of the cover designer and has been associated with WHO. The other clues are not fully confirmed, but they are plausible.
⟨/think⟩

**Final answer:** | Cristina Ortiz |

---

Table 13: An example on **BrowseComp** dataset answered by RAG-R1, with special symbols used in the search queries and search results.

| Example from BrowseComp Dataset |
| --- |

**Question:**
There was a global report released by the World Health Organisation after 2011 and before 2020 in which the forward was written by an Eritrean person who had attained a Master of Science (MSc) in Immunology of Infectious Diseases from a UK University. The introduction was written by someone whose first name began with Z and who wrote an opinion piece with a person whose first name began with the initials J, published after 2020 but before 2024. The cover was designed by a graphic artist who had completed a Leadership Strategies in Book Publishing somewhere after 2010 and before 2020 and had attained a Bachelor 2019s Degree in Graphic Design after 2000 and before 2010. This designer had previously worked at Malaria Consortium and Ogilvy & Mather. What is the first and last name of the cover designer of this report?

**Answer:** Cristina Ortiz

| Model Output |
| --- |

⟨think⟩
To answer this question, I need to identify the cover designer based on the provided details. Let's break down the information:
1. The cover designer had a Bachelor's Degree in Graphic Design between 2000 and 2010.
2. They also completed a Leadership Strategies in Book Publishing course between 2010 and 2020.
3. They worked at Malaria Consortium and Ogilvy & Mather.
I need to find a person who fits all these criteria. I will start by searching for individuals who have both a Bachelor's Degree in Graphic Design and have worked at Malaria Consortium and Ogilvy & Mather.
⟨/think⟩

<search>
graphic designer with a bachelor's in graphic design and worked at malaria consortium and ogilvy & mather ,

leadership strategies in book publishing course after 2010 before 2020
</search>

<information>
[{'title': 'Careers - Ogilvy', 'url': 'https://www.ogilvy.com/careers', 'snippet': 'We are laser focused on providing best-in-class learning and development as well as career growth opportunities for all staff.' ...
</search>

⟨think⟩
From the search results, I did not find a direct match for a graphic designer who fits all the specific criteria mentioned. However, I need to **narrow down further to find the exact person. I will try another search focusing on the intersection of the graphic designer's background and the specific roles they held**.
⟨/think⟩

<search>
graphic designer with bachelor's in graphic design and worked at malaria consortium and ogilvy & mather , # Duplicate query

name starting with z and j co-author after 2020
</search>

<information>
[{'title': 'Careers - Ogilvy', 'url': 'https://www.ogilvy.com/careers', 'snippet': 'We are laser focused on providing best-in-class learning and development as well as career growth opportunities for all staff.' ... } ...
</information>

⟨think⟩
The search did not yield a direct match for the specific individual. However, I need to refine my search to focus more on the intersection of the graphic designer's background and the specific roles they held, particularly the forward written by an Eritrean person with a Master of Science in Immunology of Infectious Diseases from a UK University.
Given the constraints, I will try one last search focusing on the Eritrean individual and the forward they wrote for the WHO report.
⟨/think⟩
... (details omitted)
⟨think⟩
Unfortunately, the search did not provide a direct match for the cover designer. However, based on the constraints and the information available, the most likely candidate is a graphic designer named Revekka Vital, as she has worked at Malaria Consortium and designed covers for WHO reports. Therefore, the first and last name of the cover designer is:
⟨/think⟩

<answer> **Rebekka Vital** </answer>

Table 14: An example on **BrowseComp** dataset answered by DeepResearcher, with special symbols used in the search queries and search results.

---

**Example from BrowseComp Dataset**

**Question:**
There was a global report released by the World Health Organisation after 2011 and before 2020 in which the forward was written by an Eritrean person who had attained a Master of Science (MSc) in Immunology of Infectious Diseases from a UK University. The introduction was written by someone whose first name began with Z and who wrote an opinion piece with a person whose first name began with the initials J, published after 2020 but before 2024. The cover was designed by a graphic artist who had completed a Leadership Strategies in Book Publishing somewhere after 2010 and before 2020 and had attained a Bachelor 2019s Degree in Graphic Design after 2000 and before 2010. This designer had previously worked at Malaria Consortium and Ogilvy & Mather. What is the first and last name of the cover designer of this report?

**Answer:** Cristina Ortiz

---

**Model Output by DeepResearcher**

⟨think⟩
First, I need to find the specific report that satisfies all the given conditions. I will start by searching for MSc and Master of Science Immunology of Infectious Diseases in the context of Eritrean individuals and their involvement with WHO reports.
Then I will look for reports with a forward from Eritrean individuals, published between 2011 and 2020. After finding the potential report, I will check for other details about the other authors and the cover designer. Let's start with the first search.
⟨/think⟩

<tool_call>
{"name": "web_search", "arguments": {"query": [ "Eritrean WHO report forward MSc Immunology of Infectious Diseases 2011-2020" ]}}
</tool_call>

<tool_response>
[{ 'search_query': 'Eritrean WHO report forward MSc Immunology of Infectious Diseases 2011-2020',
'web_page_info_list': [{ 'quick_summary': 'Lethal infectious diseases as inborn errors of immunity: toward a synthesis of the germ and genetic theories.', 'title': 'Lethal Infectious Diseases as Inborn Errors of Immunity', 'url': '...'}]}] </tool_response>

⟨think⟩
The first title "Lethal Infectious Diseases as Inborn Errors of Immunity" doesn't directly indicate it's a WHO report ...\n ... **The webpage does not provide any direct information related to the query.** However, the title and context may still be useful to cross-reference with other sources. However, to avoid making assumptions, I should search for the specific details about who authored the forward and what the cover designer's background is based on the information given in the prompt.
Let's now search for details about the author who wrote the forward and the cover designer with the given criteria.
⟨/think⟩

<tool_call>
{"name": "web_search, "arguments": {"query": [ "Eritrean WHO report forward Immunology 2011-2020" ,
"WHO report forward Immunology Master of Science Eritrean 2011-2020" ,
"WHO report graphic designer Master of Science Immunology Eritrean 2010-2020" ,
"Eritrean graphic designer Leadership Strategies in Book Publishing 2010-2020" ,
"graphic designer Eritrean Leadership Strategies in Book Publishing Malaria Consortium Ogilvy & Mather" ]}}
</tool_call>

<tool_response>
[{'search_query': 'Eritrean WHO report forward Immunology 2011-2020', 'web_page_info_list': [{'quick_summary': 'The vision for the Decade of Vaccines (2011–2020) is of a world in which...', 'title': '[PDF] Global Vaccine Action Plan - WHO', ... }]}]
</tool_response>

⟨think⟩
**Despite several searches, I haven't found any explicit information about the graphic designer of the report.** However, the most promising leads have been Simon Gebremeskel for the forward and the designer being from Eritrea with a background at Malaria Consortium and Ogilvy & Mather. Given the constraints and the information we gather, **the graphic designer is most likely working behind the scenes,** but we need to match the names with the design roles and criteria. Since Simon Gebremeskel's forward writing doesn't usually include cover design, the designer mentioned must be a separate individual whose background fits the criteria. Let's infer from the name mentioned and typical naming conventions in such industries. Considering the information about a design career with a degree after 2000 and roles at Malaria Consortium and Ogilvy & Mather, we can assume the designer's first name starting with Z and being collaborative with someone named J for the introduction. Given the limitations, the most reasonable design follows the given criteria. Assuming the cover designer is a professional matching these details: The designer's first and last name is Zeinab Mekonnen. **We match Zeinab based on the name convention following Eritrean naming patterns and fulfilling all background requirements.**
⟨/think⟩

32

<answer> **Zeinab Mekonnen** </answer>

Table 15: An example on **BrowseComp** dataset answered by Search-o1, with special symbols used in the search queries and search results.

| Example from BrowseComp Dataset |
|---|

**Question:**
There was a global report released by the World Health Organisation after 2011 and before 2020 in which the forward was written by an Eritrean person who had attained a Master of Science (MSc) in Immunology of Infectious Diseases from a UK University. The introduction was written by someone whose first name began with Z and who wrote an opinion piece with a person whose first name began with the initials J, published after 2020 but before 2024. The cover was designed by a graphic artist who had completed a Leadership Strategies in Book Publishing somewhere after 2010 and before 2020 and had attained a Bachelor 2019s Degree in Graphic Design after 2000 and before 2010. This designer had previously worked at Malaria Consortium and Ogilvy & Mather. What is the first and last name of the cover designer of this report?
**Answer:** Cristina Ortiz

| Model Output by Search-o1 |
|---|

⟨think⟩

Okay, let's try to figure out this question step by step. The question is about identifying the cover designer of a WHO report. The clues are a bit complex, so I need to break them down.

First, the report was released by WHO between 2011 and 2020. The forward (probably the foreword) was written by an Eritrean person with an MSc in Immunology of Infectious Diseases from a UK university. Then, the introduction was written by someone whose first name starts with Z. This person also wrote an opinion piece with someone whose first name starts with J, published between 2020 and 2024. The cover designer has a Bachelor's in Graphic Design (earned between 2000-2010), worked at Malaria Consortium and Ogilvy & Mather, and completed a Leadership Strategies in Book Publishing between 2010-2020.

So, the main goal is to find the cover designer's full name. Let's tackle each part.

Starting with the cover designer. The key points here are:
1. Bachelor's in Graphic Design between 2000-2010.
2. Leadership Strategies in Book Publishing between 2010-2020.
3. Worked at Malaria Consortium and Ogilvy & Mather.

I need to find someone who fits these educational and professional backgrounds. Let me think about possible candidates. Maybe someone known for working in those companies and having those degrees. But since I don't have prior knowledge, I might need to search for these details.

Wait, the question mentions that the cover designer had a Bachelor's Degree in Graphic Design after 2000 and before 2010. Also, they completed Leadership Strategies in Book Publishing after 2010 and before 2020. Let me check if there are any notable graphic designers who fit this timeline and those companies. First, let me consider Ogilvy & Mather. That's a major advertising agency. Malaria Consortium is an NGO focused on malaria. So the designer might have worked in both sectors. **Let me try to search for people who worked at both Ogilvy & Mather and Malaria Consortium.** Maybe someone with a background in graphic design and book publishing. Alternatively, maybe the name is known in the field. Let me think of possible names. Wait, the question mentions the cover designer's name is needed. Let me try to find someone with those qualifications. **But since I can't search the internet directly, I need to think if there's any famous person who fits. Alternatively, maybe the answer is a known figure. Let me think of people who have worked at Ogilvy & Mather and Malaria Consortium.**

...\n ...

**Maybe the answer is someone like "Zahra" with a last name that's common.** Alternatively, perhaps the answer is "Zahra" and the last name is "Mistry". But without specific information, I'm stuck. Maybe I need to think of possible names. Alternatively, perhaps the answer is "Zahra" and the last name is "Ali". Alternatively, maybe the answer is "Zahra" and the last name is "Khan". But I need to make sure. **Alternatively, maybe the answer is "Zahra" and the last name is "Mistry".** Alternatively, perhaps the answer is "Zahra" and the last name is "Siddiqui"...

But I'm not certain. Alternatively, maybe the answer is "Zahra" and the last name is "Ali". But I think I need to proceed with the best possible answer based on the given information.

⟨/think⟩

...\n ...

**A well-known individual who fits these criteria is \*\*Zahra Mistry\*\*.** She is recognized for her work in graphic design and publishing, and she has worked at both Malaria Consortium and Ogilvy & Mather.

—

### **Final Answer**
**Zahra Mistry**

Table 16: An example on **FRAMES** dataset answered by ours, with special symbols used in the search queries and search results.

---

**Example from FRAMES Dataset**

**Question:** Out of all of the feature-length theatrical films that John Carpenter directed before 2015, which has the longest running time?

**Answer:** Starman (1984)

---

**Model Output by Ours**

⟨think⟩

Let's tackle this question step by step. The goal is to determine which of John Carpenter's feature-length theatrical films before 2015 has the longest running time. First, I need to identify all of his feature films released before 2015. Then, I'll check their running times and find the longest one. **I'll start by searching for a list of John Carpenter's feature films released before 2015.**

⟨/think⟩

⟨|begin_search_queries|⟩

John Carpenter's feature films before 2015;

List of John Carpenter movies released before 2015

⟨|end_search_queries|⟩

⟨|begin_search_results|⟩

John Carpenter's feature films before 2015: John Carpenter's feature films directed before 2015 include the following, based on verified sources:1. **Dark Star** (1974) ...\n ...

All listed films were directed by Carpenter and released prior to 2015.

⟨|end_search_results|⟩

⟨think⟩

Now that I have a list of John Carpenter's **feature films before 2015, the next step is to determine the running time of each film and identify the longest one.** I'll search for the running times of these films.

⟨/think⟩

⟨|begin_search_queries|⟩

Running time of John Carpenter's Dark Star;

Running time of Assault on Precinct 13;

...\n ... (Parallel Search for 12 Queries)

Running time of Village of the Damned (1995);

Running time of Ghosts of Mars (2001)

⟨|end_search_queries|⟩

⟨|begin_search_results|⟩

Running time of John Carpenter's Dark Star: The running time of John Carpenter's *Dark Star* is **83 minutes**.

Running time of Assault on Precinct 13: The running time of the 1976 film *Assault on Precinct 13* is **91 minutes**.

...\n ...

Running time of Village of the Damned (1995): The running time of *Village of the Damned* (1995) is **98 minutes**, as stated on its Wikipedia page and confirmed by Variety and Nitehawk Cinema.

Running time of Ghosts of Mars (2001): The running time of *Ghosts of Mars* (2001) is **98 minutes**.

⟨|end_search_results|⟩

⟨think⟩

Now that I have the running times of all the films, I can compare them to find the longest one. Let's list them out again for clarity:

1. **Dark Star** (1974) 83 minutes

...\n ...

12. **Ghosts of Mars** (2001) 98 minutes

From this list, **the longest running time is **115 minutes** for *Starman* (1984).**

⟨/think⟩

**Final answer**: Starman (1984)

---

Table 17: An example on **FRAMES** dataset answered by RAG-R1, with special symbols used in the search queries and search results.

| Example from FRAMES Dataset |
| --- |

**Question:** Out of all of the feature-length theatrical films that John Carpenter directed before 2015, which has the longest running time?
**Answer:** Starman (1984)

| Model Output |
| --- |

⟨think⟩
To determine which of John Carpenter's feature-length theatrical films released before 2015 has the longest running time, I need to list his films from that period and their respective running times.
⟨/think⟩

<search>
John Carpenter feature films before 2015 , John Carpenter film running times
</search>

<information>
[ {'title': 'John Carpenter - Wikipedia', 'url': 'https://en.wikipedia.org/wiki/John_Carpenter', 'snippet': 'Though he has been acknowledged as an influential filmmaker, his other productions from the 1970s and the 1980s only later came to be considered cult classics; these include Dark Star (1974), Assault on Precinct 13 (1976), The Thing (1982), Christine (1983), . . .
</search>

⟨think⟩
From the search results, I can see the running times of several of John Carpenter's films. However, **the exact running times for all his films before 2015 are not listed.** I need to focus on the films he directed before 2015 and their specific running times.

<search>
John Carpenter films 2015 and earlier , John Carpenter film running times 2015 and earlier
</search>

<information>
[ {'John Carpenter - IMDb', 'url': 'https://www.imdb.com/name/nm0000118', 'snippet': 'After directing Dark Star (1974), he has helmed both classic horror films like Halloween (1978), The Fog (1980), and The Thing (1982), and noted ...
</information>

⟨think⟩
Based on the search results, **the longest running time among John Carpenter's feature-length theatrical films before 2015 is "Big Trouble in Little China,"** which has a running time of 1 hour and 44 minutes. ⟨/think⟩

<answer> **Big Trouble in Little China** </answer>

Table 18: An example on **FRAMES** dataset answered by DeepResearcher, with special symbols used in the search queries and search results.

| Example from FRAMES Dataset |
| --- |

**Question:** Out of all of the feature-length theatrical films that John Carpenter directed before 2015, which has the longest running time?
**Answer:** Starman (1984)

| Model Output by DeepResearcher |
| --- |

⟨think⟩
To find this information, I should first look for a complete list of John Carpenter's feature-length theatrical films directed before 2015 and their respective running times. Then I can find the one with the longest running time.
⟨/think⟩

<tool_call>
{"name": "web_search, "arguments": {"query": [ "list of John Carpenter films before 2015" ,

"running times of John Carpenter films before 2015" , ]}}
</tool/_call>

<tool_response>
[{ 'search_query': 'list of John Carpenter films before 2015', 'web_page_info_list': [{'quick_summary': 'John Carpenter Filmography', 'title': '...', 'url': '...'}, {'quick_summary': 'This is a list of films directed by John Carpenter that grossed...',
</tool_response>

⟨think⟩
I will first browse the webpages that are most likely to contain a list of John Carpenter's films before 2015 and their running times.
⟨/think⟩

...\n ...(Web Browsing)

⟨think⟩
**Wikipedia does not directly provide the running times of John Carpenter's films**, but it does list his filmography. I will search for more specific information regarding the running times of his films from before 2015.
⟨/think⟩

<tool_call>
{"name": "web_search, "arguments": {"query": [ "running times of John Carpenter films directed before 2015" ]}}
</tool/_call>

<tool_response>
[{ 'search_query': 'running times of John Carpenter films directed before 2015', 'web_page_info_list': []}]
</tool_response>

⟨think⟩
**I couldn't find a direct list of the running times of John Carpenter's films before 2015.** However, based on my prior searches, "The Thing" might have the longest running time as it is a noted classic horror film. I'll need to check its specific running time.
⟨/think⟩

<tool_call>
{"name": "web_search, "arguments": {"query": [ "running time of The Thing directed by John Carpente" ]}}
</tool/_call>

<tool_response>
[{ 'search_query': 'running time of The Thing directed by John Carpenter', 'web_page_info_list': [...]}]
</tool_response>

⟨think⟩
Most of the sources I found indicate that "The Thing" has a running time of approximately 109 minutes. Since there weren't any specific mentions of other films with longer running times, **it's likely that "The Thing" had the longest running time among John Carpenter's films before 2015.**
⟨/think⟩

<answer> **The Thing** </answer>

Table 19: An example on **FRAMES** dataset answered by Search-o1, with special symbols used in the search queries and search results.

---

**Example from FRAMES Dataset**

---

**Question:** Out of all of the feature-length theatrical films that John Carpenter directed before 2015, which has the longest running time?

**Answer:** Starman (1984)

---

**Model Output by Search-o1**

---

⟨think⟩

Okay, so I need to figure out which of John Carpenter's feature-length theatrical films before 2015 has the longest running time. Let me start by recalling the films he directed. John Carpenter is known for movies like Halloween, The Thing, Escape from New York, Halloween II, and others. But I need to list all his feature films before 2015 and check their running times.

...\n ...

So first, I need to list all John Carpenter's feature films before 2015. Let me start by searching for his filmography up to 2015.

⟨/think⟩

⟨|begin_search_query|⟩ John Carpenter filmography before 2015 ⟨|end_search_query|⟩

⟨|begin_search_result|⟩ John Carpenter's notable films before 2015 include: 1. **Dark Star** (1974) 2. **Assault on Precinct 13** (1976) 3. **Halloween** (1978)... ⟨|end_search_result|⟩

⟨|begin_search_query|⟩ Running time of The Thing (1982) John Carpenter ⟨|end_search_query|⟩

⟨|begin_search_result|⟩ The running time of John Carpenter's *The Thing* (1982) is **109 minutes**, as confirmed by both the Wikipedia page for the film and a review on cmsaunders.wordpress.com... ⟨|end_search_result|⟩

⟨|begin_search_query|⟩ Longest John Carpenter film before 2015 ⟨|end_search_query|⟩

⟨|begin_search_result|⟩ The longest John Carpenter film before 2015 is **Elvis (1979)**, with a runtime of **168 minutes**. This information is confirmed by Web Page 3, which identifies it as a TV film... ⟨|end_search_result|⟩

⟨|begin_search_query|⟩ Elvis (film) runtime ⟨|end_search_query|⟩

⟨|begin_search_result|⟩ The runtime of the 2022 film *Elvis*, directed by Baz Luhrmann, is **159 minutes** (2 hours and 39 minutes). This is confirmed by multiple reliable sources, including its Wikipedia page, IMDb... ⟨|end_search_result|⟩

⟨|begin_search_query|⟩ John Carpenter's Elvis film runtime ⟨|end_search_query|⟩

⟨|begin_search_result|⟩ The runtime of John Carpenter's *Elvis* (1979) is **168 minutes (2 hours and 48 minutes)** as confirmed by its IMDb page (Web Page 1) and... ⟨|end_search_result|⟩

⟨|begin_search_query|⟩ John Carpenter's filmography with runtimes ⟨|end_search_query|⟩

⟨|begin_search_result|⟩ John Carpenter's filmography with confirmed runtimes (before 2015) includes the following films, based on reliable sources like IMDb, Wikipedia, and reviews: 1. **Dark Star** (1974) **83 minutes** (1h 23m)... ⟨|end_search_result|⟩

⟨|begin_search_query|⟩ Longest John Carpenter film before 2015 ⟨|end_search_query|⟩

⟨|begin_search_result|⟩ You have searched this query. Please refer to previous results. ⟨|end_search_result|⟩

**After analyzing the search results, the longest running time among John Carpenter's feature-length theatrical films before 2015 is **Elvis (1979)****, with a runtime of **168 minutes** (2 hours and 48 minutes). While *The Thing* (1982) is often cited as one of his longer films at 109 minutes, *Elvis* clearly surpasses it in duration. This conclusion is supported by multiple reliable sources, including IMDb and JustWatch, which confirm the runtime of *Elvis* as the longest in Carpenter's filmography prior to 2015.

---

