# OpenReview forum: "Hybrid Deep Searcher: Scalable Parallel and Sequential Search Reasoning"
_ICLR.cc/2026/Conference — ICLR 2026 Poster_

### Official Review · Reviewer_RKvi · 2025-10-28

**Soundness:** 2
**Presentation:** 3
**Contribution:** 3
**Rating:** 6
**Confidence:** 5

**Summary:**

This paper focuses on multi-turn search agents. Most existing search agents perform complex information retrieval using sequential querying, whereas approaches that generate independent queries simultaneously may limit deeper, sequential reasoning. The authors construct a dataset that integrates parallel search with sequential search reasoning, collect answer trajectories, and fine-tune an LLM as a search agent via SFT.

Contributions:
* HDS-QA Dataset: An automatically constructed synthetic dataset derived from Natural Questions, containing questions that require both parallel and sequential search reasoning.
* HybridDeepSearcher: A search agent (fine-tuned Qwen3-8B) that adaptively integrates both parallel and sequential search strategies during the reasoning process.

**Strengths:**

* **Well-Motivated Problem**. The paper identifies a limitation in current search agents: sequential search agents cover limited information, while parallel search agents may sacrifice deeper, sequential reasoning capabilities.
* **Scalable Dataset Construction**. The HDS-QA dataset generation pipeline is fully automatic, eliminating the need for human annotation and enabling scalable data creation.
* **Strong Empirical Results**. The model demonstrates substantial improvements across multiple benchmarks, validating the effectiveness of the proposed approach.

**Weaknesses:**

1. **Training Methodology Limitations**. The paper collects trajectories using Qwen3-32B and fine-tunes Qwen3-8B via SFT. However, given that competitive baselines (Search-R1, RAG-R1) employ reinforcement learning for training, the absence of an RL-trained version limits understanding of the effectiveness of the method.
2. **Computational Cost and Fairness Concerns**. The use of Qwen3-32B for document summarization adds significant computational overhead without clear justification. Why not use retrieval results directly? More critically, this creates an unfair comparison with RAG-R1, the strongest baseline, which operates without a summarization module. The added computational budget makes it difficult to isolate the true benefits of the proposed approach.
3. **Dataset Construction Validity**. Based on the dataset generation process (Section 3.1) and the prompts in the appendix, the Qwen3-32B receives a bunch of extracted information and constructs the question in Step 3. Since there are no constraints, we can not determine if a generated question is really a parallel question. Furthermore, for hybrid questions structured as A|B->C (where A and B are parallel subquestions and C depends on their answers), the pipeline does not prevent information leakage. Agents may obtain sub-answers from a single branch without genuinely solving the parallel components. This raises concerns about whether the dataset truly improves parallel search capabilities or merely teaches pattern matching.
4. **Incomplete Qualitative Analysis**. The qualitative case studies (Appendix) lack comparisons with RAG-R1, the most competitive multi-query baseline, making it difficult to assess the specific advantages of HybridDeepSearcher over existing parallel query methods.

**Questions:**

1. **Formatting and Technical Errors**. The manuscript contains multiple formatting issues that affect readability: incorrect citation format at Line 048, improper vspace usage at Line 443, layout inconsistencies on Page 15, and reference errors in Appendix D. These should be addressed for publication quality.
2. **Confusing notation**. In Table 2, the original Qwen3-8B is replaced. Using "+Qwen2.5-7b" here is confusing.
3. **Parallel Query Generation Patterns**. Could the authors clarify the parallel query generation behavior observed across different case studies? In Tables 4 and 10 (MuSiQue and FRAMES examples), the first search turn generates queries that appear to rephrase the original question with minor variations, while Table 7 (BrowseComp example) produces genuinely diverse parallel queries. Since these use the same model and prompts, what factors determine when the model generates reformulations versus diverse parallel queries? Understanding this would help assess whether the model has learned systematic parallel search strategies or if the behavior is primarily question-dependent.

---

> ### Author Response · Authors · 2025-11-21
> **Response to Weakness**
>
> We sincerely appreciate the reveiwer's valuable comments.
>
> &nbsp;
>
> ---
>
> **(W1) Training Methodology Limitations: The absence of an RL-trained version limits understanding of the effectiveness of the method**
>
> We agree that applying RL could potentially yield further improvements. However, we already demonstrate that parallel-sequential search reasoning benefits under SFT. **Our main goal is to evaluate the effect of the proposed hybrid search behavior in isolation, rather than to optimize performance through a particular training algorithm such as RL.**
>
> Importantly, even under this controlled SFT-only setting, our model consistently outperforms RL-tuned baselines (DeepResearcher and RAG-R1). This suggests that the dominant factor influencing performance is the availability of explicit supervision on when to branch in parallel and when to perform sequential reasoning rather than the choice of training algorithm. We agree that applying RL on top of our model is a promising next step, and we expect it to further enhance performance.
>
> &nbsp;
>
> ---
>
> **(W2) Concerns about high computational cost caused by the 32B summarizer and the fairness of comparing it with RAG-R1, which uses no summarizer**
>
> We appreciate the reviewer’s insightful comments. Following Search-o1, we use a summarizer to reduce noise in raw web results, but we agree that using a 32B model raises fairness concerns when compared to RAG-R1, which performs no summarization.
>
> To address this directly, we ran an ablation that removes the summarizer so that our model receives the same type of input as RAG-R1. The results are shown below. **Even without a summarizer, our model consistently outperforms RAG-R1 across all datasets**, indicating that the gains are not dependent on the additional computation of the 32B summarizer. We have added the rationale for using the summarizer (L243–245) and included this ablation in Appendix C.
>
> | Setting                                       | MuSiQue | FanOutQA | FRAMES | MedBrowseComp | BrowseComp |
> |------------------------------------------|---------------|-----------------|--------------|----------------------------|-------------------|
> | RAG-R1 (baseline)                   | 32.4          | 10.0            | 45.6       | 28.2                           | 2.0                 |
> | **Ours (w/o summarizer)** | 34.3          | 15.8            | 51.1      | **33.4**                   |12.0                 |
> | **Ours (32B)**                        | **35.1**       | **20.0**    | **54.0**        | 30.4                           | **16.0**      |
>
>
> &nbsp;
>
> ---
>
> **(W3) Concerns about the lack of explicit constraints ensuring subquestion independence and preventing information leakage in hybrid questions (A|B → C)**
>
> We clarify that our data construction includes several explicit safeguards to ensure both independence and leakage prevention.
>
> **Independence of A and B.**
> (1) **Each subquestion is sourced from a different Google People-Also-Ask query, and we require their top-k retrieved documents to come from distinct sources**, ensuring that the evidence for A and B arises from disjoint information clusters.
> (2) In our complex-question prompt (page 18), we instruct the model to **“identify unique characteristics or context from these clues that indirectly lead to the given entity,** encouraging the A and B to rely on distinct semantic attributes.
>
> **Leakage prevention.**
> (3) The prompt also instructs: **“Avoid using pronouns or names in the clues that are highly related to the given entity,”** preventing overly direct or entity-revealing hints (A → C or B → C).
> (4) Finally, **we verify that neither the parallel nor the hybrid question can be solved by a naive RAG baseline** (L209–210), ensuring that no single branch is sufficient to answer the final query.
>
> Collectively, these constraints ensure that A and B remain structurally and semantically independent, making the resulting hybrid questions (A|B → C) truly require integration of both branches.
>
>
> &nbsp;
>
> ---
>
> **(W4) Qualitative studies do not include RAG-R1**
>
> We have added qualitative examples of RAG-R1 in Appendix E. Across these cases, RAG-R1 consistently struggles with coordinating multi-query search. **Its parallel queries are typically limited to two, and it often parallelizes steps that should be executed sequentially.** For example, in Table 7, it directly asks for the opening date of the first mosque in Palau without first identifying which mosque this refers to, skipping a necessary intermediate step. In Tables 10 and 14, when the initial query fails, RAG-R1 issues nearly identical queries repeatedly rather than adapting its search strategy.
>
> These patterns suggest that, even with RL, learning effective parallel–sequential query planning remains challenging. In contrast, HybridDeepSearcher explicitly models this integration, avoiding such mis-parallelization and enabling broader parallel queries.

---

> > ### Comment · Reviewer_RKvi · 2025-11-26
> >
> > Thank you for your rebuttal. I was reviewing your response to W2. However, I noticed a discrepancy between the results reported in Table 5 (Appendix C.2) and those provided in this rebuttal, e.g., 51.1 vs. 50.3 on FRAMES. **I kindly ask the authors to verify these numbers and ensure consistency across all reported results.**
> >
> > Additionally, the three variants (vanilla, w/ 8B, and w/o summarizer) exhibit inconsistent performance trends across different datasets, and employing a larger summarizer does not yield consistent improvements. I would appreciate it if the authors could provide an analysis or explanation for this observation.

---

> > > ### Author Response · Authors · 2025-11-26
> > > **Response to Comments**
> > >
> > > We appreciate the reviewer's careful reading and for pointing out the discrepancy. The number in Appendix C.2 contained a typo. We have corrected this and uploaded a revised version of the paper. We apologize for the inconvenience.
> > >
> > > Regarding the performance trend of the three variants, in our current setup, the summarizer is not jointly with the solver. **It therefore acts as a lossy intermediate model, and what constitutes "good" loss may depend on the benchmark.**
> > >
> > > * **MedBrowseComp**: This benchmark contains domain-specific medical questions that rely on rare, fine-grained cues. Both the 8B and 32B summarizers tend to drop some of this information during compression, so the full, unsummarized traces work best.
> > >
> > > * **General behavior of 8B vs 32B (other benchmarks)**.  On the remaining benchmarks, we observe a consistent qualitative pattern:
> > >   * The **8B summarizer** is more aggressive and prior-driven, sometimes dropping constraints when compressing retrievals.
> > >   * The **32B summarizer** is more cautious and literal, and tends to preserve more of the retrieved information.
> > >
> > > Concretely,
> > > * **MuSiQue & FRAMES.** The 8B summarizer is willing to “fill in” missing hops and jump to well-known answers or rounded numeric estimates, while the 32B summarizer often refuses to make these leaps and stays closer to the retrieved evidence.
> > > * **FanOutQA & BrowseComp (constraint-heavy).** Here, every constraint in the question matters. The 8B summarizer sometimes drops or distorts constraints during compression, whereas the 32B summarizer tends to preserve the full structure of the reasoning. This makes 32B huge improvements on compositional, multi-constraint tasks and explains why the larger summarizer helps more on these datasets.
> > >
> > > Overall, our results indicate that a larger summarizer does not always yield better performance. This is expected, since our method does not jointly optimize the summarizer and the solver. **Nevertheless, these experimental results suggest that our method does not rely on a heavy summarizer.**

---

> > > > ### Comment · Reviewer_RKvi · 2025-11-27
> > > >
> > > > Thanks for your response. I'll keep my positive score. Please note that Table 5 still contains an error. w/ 8B Summarizer on MuSiQue should be marked by \textbf.

---

> > > > > ### Author Response · Authors · 2025-12-03
> > > > > **Response to Comments**
> > > > >
> > > > > We thank the reviewer for the constructive feedback and helpful suggestions. We have corrected the missing bold markup in Table 5 as suggested.
> > > > > &nbsp;
> > > > >
> > > > > ---
> > > > >
> > > > > **(W1) Training Methodology Limitations: The absence of an RL-trained version limits understanding of the effectiveness of the method**
> > > > >
> > > > > To further address this concern, we also apply Group Relative Policy Optimization (GRPO) [1] to HybridDeepSearcher. As shown below, **GRPO provides modest accuracy gains, but it also increases search depth, resulting in lower AUC across benchmarks**. These findings indicate that although extended search trajectories yield modest performance improvements, the gains are slight relative to the additional search depth, resulting in lower AUC, an overall measure of search efficiency. We have updated the main paper to include these results and details; please refer to Sec. 6.
> > > > >
> > > > > &nbsp;
> > > > >
> > > > > &nbsp;&nbsp;&nbsp;&nbsp;&nbsp;&nbsp;&nbsp;&nbsp;&nbsp;&nbsp;&nbsp;&nbsp;&nbsp;&nbsp;&nbsp;&nbsp;&nbsp;&nbsp;&nbsp;&nbsp;&nbsp;&nbsp;&nbsp;&nbsp;&nbsp;&nbsp;&nbsp;&nbsp;&nbsp;&nbsp;&nbsp;&nbsp;&nbsp;&nbsp;&nbsp;**MuSiQue** &nbsp;&nbsp;&nbsp;&nbsp;&nbsp;&nbsp;&nbsp;&nbsp;&nbsp;&nbsp;&nbsp;&nbsp; **FanOutQA** &nbsp;&nbsp;&nbsp;&nbsp;&nbsp;&nbsp;&nbsp;&nbsp;&nbsp; **FRAMES**
> > > > >
> > > > > | Model          | Acc | Turn | AUC |   | Acc | Turn | AUC |   | Acc | Turn | AUC |
> > > > > |----------------|-----:|:----:|:----|:-:|-----:|:----:|:----|:-:|-----:|:----:|:----|
> > > > > | **Ours**       | 35.1 | **3.3**  | **0.30** |   | 20.0 | **3.1**  | **0.15** |   | 54.0 | **3.4**  | **0.44** |
> > > > > | **Ours (GRPO)**| **36.3** | 4.0 | 0.27 |   | **20.9** | 4.3 | **0.15** |   | **57.2** | 4.1 | 0.42 |
> > > > >
> > > > > &nbsp;
> > > > >
> > > > > &nbsp;&nbsp;&nbsp;&nbsp;&nbsp;&nbsp;&nbsp;&nbsp;&nbsp;&nbsp;&nbsp;&nbsp;&nbsp;&nbsp;&nbsp;&nbsp;&nbsp;&nbsp;&nbsp;&nbsp;&nbsp;&nbsp;&nbsp;**MedBrowseComp** &nbsp;&nbsp;&nbsp;&nbsp;**BrowseComp-50**
> > > > >
> > > > > | Model          | Acc | Turn | AUC |   | Acc | Turn | AUC |
> > > > > |----------------|-----:|:----:|:-----|:-:|-----:|:----:|:-----|
> > > > > | **Ours**       | 30.4 | **3.4** | **0.26** |   | **16.0** | **5.7** | **0.11** |
> > > > > | **Ours (GRPO)**| **31.1** | 4.1 | 0.23 |   | **16.0** | 6.4 | 0.09 |
> > > > >
> > > > > &nbsp;
> > > > >
> > > > > [1] Shao et al., "DeepSeekMath: Pushing the Limits of Mathematical Reasoning in Open Language Models", arXiv 2024.

---

> ### Author Response · Authors · 2025-11-21
> **Response to Questions**
>
> &nbsp;
>
> **(Q1, 2) Formatting and Technical Errors. & Confusing notation**
> We appreciate the reviewer for pointing these out. We have updated these to the new version.
>
> &nbsp;
>
> ---
>
> **(Q3) Could the authors clarify the parallel query generation behavior observed across different case studies?**
>
> We appreciate the reviewer’s careful examination of the case studies in Tables 6, 9, and 13. Parallel queries in our setting serve two roles: (i) **retrieving complementary information when the task decomposes into independent subquestions**, and (ii) **mitigating retrieval uncertainty when a single query may fail to surface the required evidence**. Importantly, both behaviors arise from the same model and prompt.
>
> In Table 9 (BrowseComp), the question naturally decomposes into independent subgoals, leading the model to generate genuinely diverse parallel queries that retrieve different entities or attributes. In contrast, the MuSiQue and FRAMES examples (Tables 6 and 13) involve tasks where **small variations of a single query can still meaningfully reduce retrieval uncertainty**. As Figure 6 shows, even Search-o1 issues repeated queries (e.g., “first mosque in Koror Palau”) after receiving incomplete or noisy results—indicating that redundancy can be necessary for reliable retrieval.
>
> In summary, **the model adapts its parallel-query strategy to the task structure**: it produces diverse queries when multiple subgoals exist, and redundant queries when doing so increases robustness against retrieval errors.

---

### Official Review · Reviewer_B25Z · 2025-10-31

**Soundness:** 3
**Presentation:** 4
**Contribution:** 2
**Rating:** 2
**Confidence:** 5

**Summary:**

This paper introduces a dataset and a training method that teaches the model to perform parallel search in addition to the common sequential search. The proposed dataset, HDS-QA, requires both parallel and sequential search, which can serve as a challenging training set that is scarce in the current training community.

**Strengths:**

1. The paper raised attention to parallel search, which is an under-studied capability of deep research models.
2. Using parallel search might enable the model with extended capability to gather information and more efficient utilization of context length.

**Weaknesses:**

1. The paper highlights its novelty on doing parallel search in addition to sequential search; however, the main change can be understood as adding a parser on the tool side to parse a list of queries from a single query string, which seems trivial to me. This has been implemented in some open-source projects like MiroMind as a small feature.
2. The construction of the dataset claims to require parallel search, but the ablation failed to accurately demonstrate how training a sequential search model on the same data might not work well as parallel search.

**Questions:**

1. There are many works from Tongyi (e.g., WebWalker, WebExplorer, etc.) and the fully open-sourced work called ASearcher on constructing synthetic BrowseComp-like QA datasets for deep research training. Could the author clarify the difference of HDS-QA from these works?
2. The choice of the 50 questions from BrowseComp needs a bit more justification. Using a super small test set will make the results high variance. Did the author benchmarked the variance and show what is the CI of this evaluation and how significant is the current gap between different models?
3. Could the author compare their models with baselines that are trained on synthetic data designed for sequential search, such as WebExplorer-8B and ASearcher-8B to show the benefit of parallel search?
4. I'm curious if the main benefit comes from the synthetic data or from developing parallel search. Could the author train the model for sequential search on the same training questions and ablate the effect? In addition, I wonder if the "parallel search" can be viewed as a new search tool that can parse a long string of queries and divide into sub-queries to gather information, because the model inference pipeline seems exactly the same as sequential search.

---

> ### Author Response · Authors · 2025-11-21
> **Response to Weakness**
>
> We sincerely appreciate the reviewer's valuable comments.
>
> &nbsp;
>
> ---
>
> **(W1) Doing a parallel search can be done by adding a parser on the tool side to parse a list of queries from a single query string (e.g., MiroMind)**
>
> We believe that a tool-side parser **without hybrid search reasoning produces ineffective and inefficient sub-queries.**
>
> In our experiments, the DeepResearcher baseline with a tool-call parser shows this failure. For the question “When did the first mosque open where the Federated States of Micronesia Maritime Boundary Treaty was signed?”, its initial sub-queries—“location of Federated States of Micronesia Maritime Boundary Treaty signing” and “when was the first mosque opened in the treaty signing location”—**retain the unresolved phrase “the treaty signing location,” merely restating the question and blocking useful evidence retrieval.**
>
> This shows that a tool-side parser is insufficient. An effective strategy must: **(i) decide whether tool use should be parallel or sequential, (ii) issue parallel queries accordingly, and (iii) integrate retrieved evidence for subsequent steps.**
> Unlike the baseline, our HybridDeepSearcher (Table 6) first identifies Majuro as the treaty-signing location and then queries for the mosque’s opening date there, demonstrating the intended hybrid (parallel-then-sequential) behavior.
>
> &nbsp;
>
> ---
>
> **(W2) The ablation failed to accurately demonstrate how training a sequential search model on the same data might not work**
>
> We appreciate the reviewer’s insightful comment. To directly test whether the gains arise from hybrid trajectories, we constructed an ablation where **the model is trained on trajectories that issue only one query per reasoning step**, keeping the questions identical while removing any parallel behavior. Using the same data-generation pipeline, this yields 1.6K valid trajectories. The results are shown below:
>
> | Model                  | Musique | FanOutQA | FRAMES | MedBrowseComp | BrowseComp |
> |------------------------|---------|----------|--------|----------------|-------------|
> | Search-o1           | 23.4  | 8.7      | 48.6   | 21.6           | 2.0         |
> | **Single-query fine-tune**  |  21.6 |  8.7  |  33.8  |   26.8  |   8.0     |
> | Ours                    | **35.1**   | **20.0**    | **54.0**  | **30.4**           | **16.0**       |
>
> These results show that **sequential-only supervision does not generalize well across diverse tasks.** Single-query fine-tuning consistently underperforms our method and even falls below Search-o1 on MuSiQue and FRAMES, suggesting that restricting trajectories to a single query can lead to overfitting.

---

> ### Author Response · Authors · 2025-11-21
> **Response to Questions**
>
> &nbsp;
>
> **(Q1) Comparison with recent (also concurrent) work with training datasets**
>
> Existing work, such as WebWalker, and the very recent concurrent works, WebExplorer and ASearcher, **focus on creating increasingly difficult long-horizon tasks to elicit multi-step sequential tool-use trajectories**. These datasets increase difficulty by expanding traversal graphs or repeatedly merging retrieved information, leading to deeper sequential reasoning chains.
>
> In contrast, **HDS-QA is not aimed at making tasks harder, but at teaching a model a hybrid search structure** that combines sequential and parallel search.
> - **On the question side**, we explicitly design tasks containing one parallel hop followed by one sequential hop, providing a controlled testbed for hybrid search.
> - **On the trajectory side**, demonstrations show the model (i) issuing parallel subqueries, (ii) aggregating retrieved evidence, and (iii) using it in a subsequent sequential search step.
>
> This offers direct supervision for parallel–sequential search structuring, which prior and concurrent long-horizon datasets do not provide. We have added these comparisons to the Related Work section (L156–159).
>
> &nbsp;
>
> ---
>
> **(Q2) Justify the 50-question subset and show statistical significance of the performance gaps.**
>
> BrowseComp requires long-horizon browsing, making full multi-step evaluation expensive. We therefore construct a practical yet challenging subset (BrowseComp-50) by selecting questions solvable by a strong search agent (OpenAI o3 Deep Research) within a 5-minute limit. Because no API is available, we manually evaluated questions in order and selected the first 50 that were solved within this time. The procedure is described in Lines 290–293.
>
> To address concerns about variance from the smaller test set, we report **Wilson confidence intervals** [4] for all models.
> - R1-Searcher: 0% (0/50), 95% CI ≈ [0.0%, 7.1%]
> - Search-o1, Search-R1, DeepResearcher, and RAG-R1: 2% (1/50), 95% CI ≈ [0.35%, 10.5%]
> - Ours: 16% (8/50), 95% CI ≈ **[8.3%, 28.5%].
>
> The lower bound of our CI (8.3%) exceeds the upper bound of R1-Searcher (7.1%), demonstrating a clear improvement even with 50 samples. Compared to the 2% baselines, our method achieves +14 percentage points, and a two-sample test yields **p ≈ 0.014 (< 0.05), confirming statistical significance**.
>
> &nbsp;
>
> ---
>
> **(Q3) Further performance comparison with concurrent work (WebExplorer and ASearcher) with web-browsing**
>
> **WebExplorer-8B and ASearcher-8B are concurrent works under the ICLR policy**, released on arXiv in August and September, respectively. In addition, **these models support not only web search but also full web-browsing tool calls, whereas our model uses only a search tool**. As a result, their training objectives and capabilities involve general web navigation rather than sequential search alone. For these reasons, a direct comparison would not be fair.
>
> &nbsp;
>
> ---
>
> **(Q4) Ablation with a sequential-only search fine-tuned model and the question of whether parallel search merely parses long queries into sub-queries.**.
>
> For the ablation, please refer to our response to **W2**, where we directly compare the sequential-only fine-tuned model with ours. Contrary to the reviewer’s view, our parallel search does far more than splitting a long output into multiple queries: the model must decide what to resolve at each step, determine which components should be decomposed into sub-queries, and integrate the retrieved evidence into subsequent reasoning (**precisely the behavior clarified in W1**). Our HDS-QA unifies these processes and enables truly hybrid parallel–sequential search.
>
> &nbsp;
>
> ---
>
>
> [1] Wu et al., “Webwalker: Benchmarking LLMs in Web Traversal”, ACL 2025.
> [2] Gao et al., “Beyond Ten Turns: Unlocking Long-Horizon Agentic Search with Large-scale Asynchronous RL”, ArXiv 2025.
> [3] Liu et al., “WebExplorer: Explore and Evolve for Training Long-horizon Web Agents”, ArXiv 2025.
> [4] E. B. Wilson, “Probable Inference, the Law of Succession, and Statistical Inference”, Journal of the American Statistical Association, 1927.

---

### Official Review · Reviewer_RTtM · 2025-11-01

**Soundness:** 3
**Presentation:** 3
**Contribution:** 2
**Rating:** 6
**Confidence:** 3

**Summary:**

The paper introduces HybridDeepSearcher, a deep search agent that integrates parallel and sequential search to perform RAG-based reasoning and question answering. To train it, the paper first synthesizes the HDS-QA dataset, focusing on generating hard, high-quality data points and corresponding parallel search traces, which consist of reason-query-retrieval. By using the HDS-QA dataset, fine-tuning alone is able to result in state-of-the-art performance on multiple challenging multi-hop QA datasets, providing both efficiency and effectiveness.

**Strengths:**

1. The paper is well-written with clear illustrations.
2. The model training recipe is clean and systematic.
3. The dataset is generated using Google’s People Also Ask feature, which seems to have significantly increased the difficulty and quality of the generated questions.
4. The experimental evaluation is comprehensive with a sufficient number of datasets and baselines. The analytical studies on test-time scaling are also insightful.

**Weaknesses:**

1. The pipeline still uses a bigger model Qwen3-32B to summarize retrieved documents, which may incur unwanted costs. Have the authors examined the performance using a smaller model, like the 8B model, as a summarizer?
2. As the primary contribution is a high-quality data synthesis pipeline, the paper did not explore the scaling behavior with respect to the number of generated data. It would be interesting to explore questions like how much data is needed, and whether more data helps.

**Questions:**

1. Generating parallel-hop questions is very interesting, but how does sequential search perform on this task compared to parallel search with the same number of queries submitted?
2. The method uses the Jina search API to perform parallel search. Are all baselines, including RL-based ones (e.g., search-R1), also trained on and use the Jina API?
3. The generated dataset can be put to better use. Does the author think that the HDS-QA dataset (without trajectories) can be used for RL training? What are the advantages and disadvantages?
4. Have the author examine the model performance against other baseline methods on one or two simpler general QA or multi-hop QA datasets used by Search-R1?
5. Typos: line 1191, Table numbers are missing.

---

> ### Author Response · Authors · 2025-11-21
> **Response to Weakness**
>
> We sincerely appreciate the reviewer’s valuable comments.
> &nbsp;
> &nbsp;
>
> **(W1) The proposed method still uses a bigger 32B summarizer, incurring unwanted costs.**
>
> To address the reviewer’s concern about using a 32B summarizer, we ran additional ablations with (i) no summarizer and (ii) an 8B summarizer. The results (ACC) are shown below:
>
> | Setting                                       | MuSiQue | FanOutQA | FRAMES | MedBrowseComp | BrowseComp |
> |------------------------------------------|---------------|-----------------|--------------|----------------------------|-------------------|
> | RAG-R1 (baseline)                   | 32.4          | 10.0            | 45.6       | 28.2                           | 2.0                 |
> | **Ours (w/o summarizer)** | 34.3          | 15.8            | 51.1      | **33.4**                   |12.0                 |
> | **Ours (8B)**                          | **37.6**  | 16.1            | **55.6**  | 29.5                        | 12.0           |
> | **Ours (32B)**                        | 35.1          | **20.0**    | 54.0        | 30.4                           | **16.0**      |
>
> These results show that a 32B summarizer is not required for strong performance. **Both the 8B and no-summarizer variants outperform the RAG-R1 baseline, with only modest drops (0.9 and 1.7 on average) compared to the 32B version**.
>
> We initially followed the Search-o1 configuration, which uses a larger summarizer to reduce noise in retrieved documents. However, our ablations demonstrate that the method remains robust even with a smaller or no summarizer. We have added these results to Appendix C.
>
> &nbsp;
>
> ---
>
> **(W2) Scaling behavior with respect to the amount of synthesized training data is underexplored**
>
> We appreciate the reviewer’s interest in the data-scaling behavior. Since our primary goal is to teach the model to coordinate sequential and parallel search, **our main experiments focus on scaling with test-time search budget (e.g., more API calls or search turns)**.
>
> For completeness, we also analyze scaling with respect to synthesized data by varying the number of sampled trajectories per question. Sampling 2 trajectories yields 1,363 trajectories from 671 questions, and sampling 4 yields 2,111 trajectories from 773 questions. The results are shown below. Although the number of questions grows only modestly, the **increased diversity from more trajectories leads to clear performance improvements**.
>
> | # Trajectory samples | MuSiQue | FanOutQA | FRAMES |
> |-----------------------------------|---------------|-----------------|--------------|
> |2 per question |33.1 |16.4 | 44.5 |
> |4 per question |**35.1** |**20.0** |**54.0** |

---

> ### Author Response · Authors · 2025-11-21
> **Response to Questions**
>
> &nbsp;
>
> **(Q1) Sequential vs parallel search with the same number of queries**
>
> We thank the reviewer for the suggestion. Due to resource constraints, we apply a fixed query budget only to HybridDeepSearcher while leaving all baselines unrestricted. As shown in **Figure 1 and Fig. 5(b), we limit HybridDeepSearcher to 2, 4, 8, and 16 queries**. We find that:
> - **With only 4 queries, our method matches baseline performance on one benchmark and surpasses it on three.**
> - **With 8 queries, it outperforms all baselines**, even though the baselines are not budget-limited.
>
> This advantage comes from the fact that parallel-hop search retrieves multiple relevant evidence pieces per query, whereas sequential pipelines acquire only one per step and must execute longer chains to cover the same information. Under the same query budget, sequential search therefore accumulates more partial or drifting context over more steps, while our parallel-hop approach mitigates compounding errors by collecting diverse evidence at each step.
>
>
> &nbsp;
>
> ---
>
> **(Q2) Are all baselines also trained on and using the Jina API?**
>
> Some baselines, including Search-R1, were not trained using the Jina web search API, which raises fairness concerns. To address this, we also report results from our model **without the summarizer. As shown in our response to W1, this variant still outperforms the baselines**, despite not being trained on search API outputs. We hope this alleviates the reviewer’s concern.
>
> &nbsp;
>
> ---
>
> **(Q3) Is it possible to do RL using the generated dataset? Advantages and disadvantages**
>
> Yes, the generated questions can be used for RL. Prior work [1,2] shows that RL can strengthen search-based agents by promoting deeper iterative search and reasoning, and we expect similar benefits for HybridDeepSearcher. **RL would also allow us to utilize HDS-QA questions without successful trajectories, which are not directly usable for supervised training**.
>
> However, RL methods such as GRPO **require substantial computational resources** and tend to be less stable than supervised approaches. For this reason, we view RL as a promising but computationally expensive direction and leave it for future work.
>
> [1] Tan et al., “RAG-R1: Incentivize the Search and Reasoning Capabilities of LLMs through Multi-query Parallelism”, arXiv 2025.
> [2] Wu et al., “WebDancer: Towards Autonomous Information Seeking Agency”, NIPS 2025.
>
> &nbsp;
>
> ---
>
> **(Q4) Have the author examined the model performance on simpler QA datasets?**
> Yes. As shown in Figure 4, we evaluate our method on **2-hop MuSiQue** to assess performance on simpler multi-hop QA. A detailed comparison with Search-R1 and other baselines is provided below. These results show that parallel search not only scales to complex settings but also delivers strong performance on simpler multi-hop QA tasks
>
> | **# Evidences** | **Search-o1** | **Search-R1** | **RAG-R1** | **Ours** |
> |-----------------|---------------|---------------|------------|----------|
> | **2**           | 42.2          | 39.5          | 42.6       | **44.6** |
> | **3**           | 24.3          | 21.3          | 25.4       | **26.6** |
> | **4**           | 15.3          | 12.9          | 15.3       | **23.5** |
>
> &nbsp;
>
> ---
>
> **(Q5) Typos in table numbers  in the Appendix**
>
> We appreciate the reviewer for catching this issue. The missing table numbers have now been added in the updated manuscript.

---

### Official Review · Reviewer_NCby · 2025-11-03

**Soundness:** 4
**Presentation:** 3
**Contribution:** 3
**Rating:** 4
**Confidence:** 4

**Summary:**

The authors of this paper introduce HDS-QA, a new training dataset that is used to teach models how to make multiple parallel and sequential web search calls for answering hard reasoning-intensive questions. The authors fine-tune a qwen3-8b model on this dataset to obtain the HybridDeepSearcher model. HybridDeepSearcher uses parallel and sequential search to obtain state-of-the art performance on multiple hard question-answering datasets.

**Strengths:**

The strengths of this paper are:
- Authors compare to multiple recent methods that serve as strong baselines.
- HybridDeepSearcher beats baselines on multiple datasets. Some of these datasets like BrowseComp are very challenging QA datasets.
- The scaling curves convincingly show that HybridDeepSearcher is able to use extra tool calls and search turns effectively.

**Weaknesses:**

The weaknesses of this paper are:
- Parallel and multi-hop questions are generated using one fixed pipeline. This might lead to limited diversity and many questions might have very similar patterns.
- Is HDS-QA only useful for improving search capabilities in 7-8b models or do smaller/larger models also benefit from training on HDS-QA? For larger models you likely need traces from a larger teacher model?
- Experiments are limited to SFT on HDS-QA and there no experiments to see if performance can further be improved by using RL.

**Questions:**

Here are a couple questions:
- Are all questions constructed by starting from a question in the NQ dataset?
- Does HybridDeepSearcher overthink on easier benchmarks? Figure 3 shows that HybridDeepSearcher is worse compared to other baselines when fewer calls are utilized. Is this a concern for easier datasets?
- How does the performance of HybridDeepSearcher scale with training data?
- What is the distribution of the number of parallel and sequential tool calls required for samples in HDS-QA?

---

> ### Author Response · Authors · 2025-11-21
> **Response to Weaknesses**
>
> We sincerely appreciate the reviewer’s valuable comments.
> &nbsp;
> &nbsp;
> **(W1) Questions are generated using one fixed pipeline, leading to limited diversity and similar patterns**
>
> It is true that our question generator uses a fixed structure with one parallel hop and one sequential hop. This design is intentionally minimal: it provides a minimal yet reliable setup that ensures both parallel and sequential reasoning are required, while avoiding noise from overly flexible templates.
>
> Despite the simple template, the resulting **HDS-QA trajectories show substantial diversity in both sequential depth and parallel width**:
> |                    | 1 | 2 | 3 | 4 | 5+ |
> | --- | ---| --- | ---| --- | --- |
> | Sequential | 586 | 1958 | 2044 | 166 | 36 |
> | Parallel       | 211 | 1319 | 424 | 119 | 38 |
>
> These statistics indicate that the fixed question form serves as a stable testbed through which the model consistently decide when parallel retrieval or sequential reasoning is appropriate.
>
> Finally, the model shows strong generalization across five benchmarks, which we attribute to the diversity of its reasoning trajectories rather than variation in question format.
>
> &nbsp;
>
> ---
>
> **(W2) Is HDS-QA applicable to larger/smaller models? Always need traces from a larger teacher?**
>
> We thank the reviewer for the thoughtful question. Our experiments below show that **HDS-QA benefits both smaller and larger models, and importantly, the teacher model does not need to be larger than the student.** (We have included this ablation study in Appendix C.)
>
> **Large model (32B)** - We fine-tune the Qwen3-32B model on HDS-QA and observe consistent improvements as shown below. Since the hybrid-search trajectories were generated by the same 32B model, these results indicate that a teacher model larger than the student is not required.
>
> | Model         |  MuSiQue |  FanoutQA | FRAMES | MedBrowseComp | BrowseComp |
> |---------------|-------|-------|---------|------|---------|
> | Search-o1 (32B) | 35.9  | 17.7    | 58.1  | 33.2  | 6.0   |
> | Ours (32B) | **37.5**  | **30.0**  | **59.8**  |**36.0**  | **18.0**  |
>
> **Small model (4B)** - We also observe clear gains for smaller models. Fine-tuning Qwen3-4B on HDS-QA leads to substantial improvements over Search-o1 with the same backbone:
>
> | Model         |  MuSiQue |  FanoutQA | FRAMES | MedBrowseComp | BrowseComp |
> |---------------|-------|-------|---------|------|---------|
> | Search-o1 (4B) | 28.5  |  8.1  |  45.2  | 19.2  |  2.0  |
> | Ours (4B) | **33.4**  | **17.7**  | **51.8**  |**30.4**  | **8.0**  |
> &nbsp;
> ---
>
> **(W3) Lack of experiments evaluating RL-based performance improvements**
>
> We appreciate the reviewer’s suggestion to examine whether RL could further improve performance. We would like to note that **our primary research question lies in the effect of integrating parallel and sequential search**. Thus, the contribution of this work is independent of whether RL algorithms are applied or not.  We agree that RL-based extensions are promising directions for future work, although we were not able to experiment with them during the rebuttal due to their significant computational cost.

---

> ### Author Response · Authors · 2025-11-21
> **Response to Questions**
>
> &nbsp;
>
> **(Q1) Are all questions constructed by starting from a question in the NQ dataset?**
> Yes, they are, as noted in L194.
>
> &nbsp;
>
> ---
>
> **(Q2) Does the model overthink? Figure 3 shows that HybridDeepSearcher is worse compared to other baselines when fewer calls are utilized, raising concerns of overthinking on easier benchmarks.**
>
> (1) **“Does HybridDeepSearcher overthink on easier benchmarks?”**
> No, **Figure 6 (Appendix B.2) shows that HybridDeepSearcher answers correctly with fewer tokens** than all baselines, indicating more efficient reasoning rather than overthinking.
>
> (2) **“Figure 3 shows that HybridDeepSearcher is worse with fewer calls.”**
> To clarify the purpose of Figure 3, **it is intended to show how models scale as additional search budget becomes available**.   Baselines saturate quickly and show limited gains with more search capacity, whereas HybridDeepSearcher continues to improve as search resources increase, which is the intended test-time scaling behavior of our approach.
>
> (3) **“Is this a concern for easier datasets?”**
> We do not believe so. The upper part of Figure 3 provides the relevant comparison: **HybridDeepSearcher achieves stronger performance with fewer search turns, reducing latency**. It surpasses most baselines within 2 turns and exceeds all baselines by 4 turns. This efficiency comes from issuing multiple parallel search queries within a single turn, allowing the model to gather substantial information early in the reasoning process.
>
> &nbsp;
>
> ---
>
> **(Q3) Does the performance of HybridDeepSearcher scale with training data?**
>
> Yes, performance improves as the amount of training data increases. To assess scalability, we compare models **trained with 2 trajectories per question** and **4 trajectories per question**. The former yields 1,363 trajectories from 671 unique questions, while the latter yields 2,111 trajectories from 773 unique questions. The results are shown below:
> | # Trajectory samples | MuSiQue | FanOutQA | FRAMES |
> |-----------------------------------|---------------|-----------------|--------------|
> |2 per question |33.1 |16.4 | 44.5 |
> |4 per question |**35.1** |**20.0** |**54.0** |
>
> These results indicate that **increasing trajectory diversity leads to larger performance gains**.
>
> &nbsp;
>
> ---
>
> **(Q4) Distribution of the number of parallel and sequential tool calls for questions in HDS-QA**.
>
> Each question in HDS-QA contains one parallel hop and then one sequential hop. The parallel hop involves 3, 4, or 5 tool calls, with rates of 38.1%, 35.9%, and 26.0%, respectively, while the sequential hop involves only 1. The variety of parallel breadth, with subsequent sequential depth, enables the dataset to cover multiple levels of parallel search complexity while maintaining a consistent sequential reasoning step.

---

> ### Author Response · Authors · 2025-12-03
> **Response to W3**
>
> **(W3) Lack of experiments evaluating RL-based performance improvements**
>
> **[Updated]** To further address this concern, we also apply GRPO [1] to HybridDeepSearcher. As shown below, **GRPO provides modest accuracy gains, but it also increases search depth, resulting in lower AUC across benchmarks**. These findings indicate that although extended search trajectories yield modest performance improvements, the gains are slight relative to the additional search depth, resulting in lower AUC, an overall measure of search efficiency. We have updated the main paper to include these results and details; please refer to Sec. 6.
>
> &nbsp;&nbsp;&nbsp;&nbsp;&nbsp;&nbsp;&nbsp;&nbsp;&nbsp;&nbsp;&nbsp;&nbsp;&nbsp;&nbsp;&nbsp;&nbsp;&nbsp;&nbsp;&nbsp;&nbsp;&nbsp;&nbsp;&nbsp;&nbsp;&nbsp;&nbsp;&nbsp;&nbsp;&nbsp;&nbsp;**MuSiQue** &nbsp;&nbsp;&nbsp;&nbsp;&nbsp;&nbsp;&nbsp;&nbsp;&nbsp;&nbsp;&nbsp;&nbsp;&nbsp;&nbsp;&nbsp;&nbsp;&nbsp;&nbsp;**FanOutQA** &nbsp;&nbsp;&nbsp;&nbsp;&nbsp;&nbsp;&nbsp;&nbsp;&nbsp;&nbsp;&nbsp;&nbsp;&nbsp; **FRAMES**&nbsp;&nbsp;&nbsp;&nbsp;&nbsp;&nbsp;&nbsp;&nbsp;&nbsp;&nbsp;&nbsp;&nbsp;&nbsp;&nbsp;&nbsp;&nbsp;&nbsp;&nbsp;&nbsp;&nbsp;&nbsp;&nbsp;**MedBC** &nbsp;&nbsp;&nbsp;&nbsp;&nbsp;&nbsp;&nbsp;&nbsp;&nbsp;&nbsp;&nbsp;&nbsp;**BrowseComp-50**
>
> | Model          | Acc | Turn | AUC |   | Acc | Turn | AUC |   | Acc | Turn | AUC | | Acc | Turn | AUC |   | Acc | Turn | AUC |
> |----------------|-----:|:----:|:----|:-:|-----:|:----:|:----|:-:|-----:|:----:|:----|:-:|-----:|:----:|:----|:-:|-----:|:----:|:----|
> | **Ours**       | 35.1 | **3.3**  | **0.30** |   | 20.0 | **3.1**  | **0.15** |   | 54.0 | **3.4**  | **0.44** || 30.4 | **3.4** | **0.26** |   | **16.0** | **5.7** | **0.11** |
> | **Ours (GRPO)**| **36.3** | 4.0 | 0.27 |   | **20.9** | 4.3 | **0.15** |   | **57.2** | 4.1 | 0.42 || **31.1** | 4.1 | 0.23 |   | **16.0** | 6.4 | 0.09 |
>
> [1] Shao et al., "DeepSeekMath: Pushing the Limits of Mathematical Reasoning in Open Language Models", arXiv 2024.

---

### Author Response · Authors · 2025-12-04
**Author Final Remarks**

Dear Reviewers and AC,

We sincerely thank the reviewers for their thoughtful evaluations and for recognizing the importance and potential impact of our work.

Across the reviews, several key strengths of our contributions were consistently acknowledged:

- **Clear Motivation**: Reviewers (RKvi, B25Z) agreed that the paper identifies an important limitation of existing search agents. Sequential search limits information coverage, and parallel search limits deeper reasoning. This motivates the need for a hybrid approach.
- **High-Quality and Scalable Dataset**: Reviewers (NCby, RTtM, RKvi) noted that HDS-QA is fully automatic, scalable, and produces more challenging questions that explicitly require both parallel and sequential reasoning.
- **Strong Empirical Results**: Reviewers (NCby, RTtM, RKvi) found that HybridDeepSearcher outperforms strong and recent baselines across multiple challenging datasets. Reviewer NCby also highlighted the breadth and rigor of our comparisons.
- **Insightful Scaling Analyses**: Reviewers (NCby, RTtM, RKvi) found the scaling-curve analyses convincing and agreed that the model makes effective use of additional search turns and both parallel and sequential tool calls.

While some concerns were raised, we believe our rebuttal has thoroughly clarified and addressed all points with detailed explanations and supporting evidence.

We again thank the reviewers and AC for their time, effort, and constructive feedback throughout the review process.

---

### Meta-Review · Area_Chair_N6UC · 2025-12-28

**Summary:**

The authors develop a new dataset to train retrieval models to do both parallel and sequential search. Reviewers had questions about the experimental setup and comparisons (e.g., isolating the effects of adding parallel search vs. having more training data, and the variance in only using 50 evaluation samples from BrowseComp). These were largely addressed in the rebuttal. I recommend that the authors incorporate the feedback into their submission: they should clarify their contributions and revise the framing accordingly, as well as expand the evaluation set so that they can draw conclusions more robustly more >50 samples. It would also be helpful to clarify the construction of the dataset -- the example seems to show a fixed template of parallel -> sequential search, whereas in their rebuttal, the authors claim that some constructed questions can have a paralle and/or sequential width of 5.

**Reviewer Concerns:**

See main comment.

**Reviewer Scores:**

4626 -> 5656

---

### Decision · Program_Chairs · 2026-01-26

Accept (Poster)